# Improved Analysis of Clipping Algorithms for Non-convex Optimization

**Bohang Zhang**[*]
Key Laboratory of Machine Perception, MOE,
School of EECS, Peking University
zhangbohang@pku.edu.cn

**Jikai Jin**[*]
School of Mathematical Sciences
Peking University
jkjin@pku.edu.cn

**Cong Fang**
University of Pennsylvania
fangcong@pku.edu.cn

**Liwei Wang**[†]
Key Laboratory of Machine Perception, MOE,
School of EECS, Peking University
Center of Data Science, Peking University
wanglw@cis.pku.edu.cn

## Abstract

Gradient clipping is commonly used in training deep neural networks partly due to its practicability in relieving the exploding gradient problem. Recently, Zhang et al. [2020a] show that clipped (stochastic) Gradient Descent (GD) converges faster than vanilla GD/SGD via introducing a new assumption called $(L_0, L_1)$-smoothness, which characterizes the violent fluctuation of gradients typically encountered in deep neural networks. However, their iteration complexities on the problem-dependent parameters are rather pessimistic, and theoretical justification of clipping combined with other crucial techniques, e.g. momentum acceleration, are still lacking. In this paper, we bridge the gap by presenting a general framework to study the clipping algorithms, which also takes momentum methods into consideration. We provide convergence analysis of the framework in both deterministic and stochastic setting, and demonstrate the tightness of our results by comparing them with existing lower bounds. Our results imply that the efficiency of clipping methods will not degenerate even in highly non-smooth regions of the landscape. Experiments confirm the superiority of clipping-based methods in deep learning tasks.

## 1   Introduction

The problem of the central interest in this paper is to minimize a general non-convex function presented below:

$$\min_{x \in \mathbb{R}^d} F(x), \tag{1}$$

where $F(x)$ can be potentially stochastic, i.e.

$$F(x) = \mathbb{E}_{\xi \sim \mathcal{D}} \left[ f(x, \xi) \right].$$

For non-convex optimization problems in form of (1), since obtaining the global minimum is NP-hard in general, this paper takes the concern on a reasonable relaxed criteria: finding an $\varepsilon$-approximate first-order stationary point such that $\|\nabla F(x)\| \le \varepsilon$.

---

[*]Equal Contributions.
[†]Corresponding author.

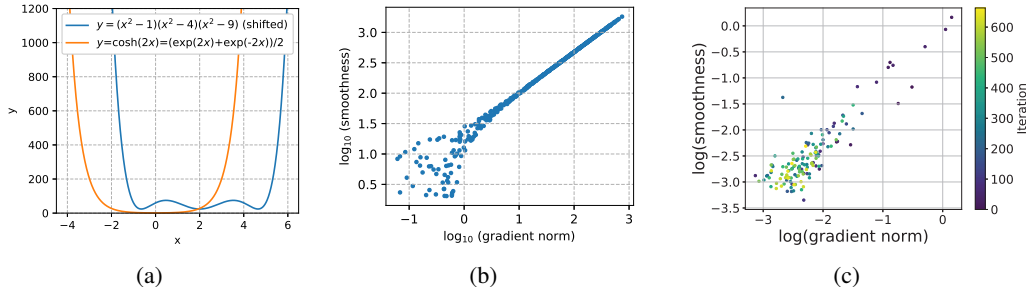

Figure 1: (a) Some simple examples of $(L_0, L_1)$-smooth functions that are not $L$-smooth. (b) The magnitude of gradient norm $\|\nabla F(x)\|$ w.r.t the local smoothness $\|\nabla^2 F(x)\|$ on some sample points for a polynomial $F(x, y) = x^2 + (y - 3x + 2)^4$. We use log-scale axis. The local smoothness strongly correlates to the gradient. (c) Gradient and smoothness in the process of LSTM training, taken from Zhang et al. [2020a].

We consider gradient-based algorithms to solve (1) and separately study two cases: i) the gradient of $F$ given a point $x$ is accessible; ii) only a stochastic estimator is accessible. We shall refer the former as the deterministic setting and the latter as the stochastic setting, and we analyze the (stochastic) gradient complexities to search an approximate first-order stationary point for Problem (1).

Gradient clipping [Pascanu et al., 2012] is a simple and commonly used trick in algorithms that adaptively choose step sizes to make optimization stable. For the task of training deep neural networks (especially for language processing tasks), it is often a standard practice and is believed to be efficient in relieving the exploding gradient problem from empirical studies [Pascanu et al., 2013]. More recently, Zhang et al. [2020a] proposed an inspiring theoretical justification on the clipping technique via introducing the $(L_0, L_1)$-smoothness assumption. The concept of $(L_0, L_1)$-smoothness is defined as follows.

**Definition 1.1** *We say that a twice differentiable function $F(x)$ is $(L_0, L_1)$-smooth, if for all $x \in \mathbb{R}^d$ we have $\|\nabla^2 F(x)\| \leq L_0 + L_1\|\nabla F(x)\|$.*

This assumption can be further relaxed such that twice differentiability is not required (see Remark 2.3). Therefore the standard $L$-smoothness assumption (i.e. the gradient of $f$ is $L$-Lipschitz continuous) is stronger than the $(L_0, L_1)$-smoothness one in the sense that the latter allows $\|\nabla^2 F(x)\|$ to have a linear growth with respect to $\|\nabla F(x)\|$.

$(L_0, L_1)$-smoothness is more realistic than $L$-smoothness. *Firstly*, it includes a variety of simple and important functions which, unfortunately, do not satisfy $L$-smoothness. For example, all univariate polynomials (which can possibly be non-convex) are $(L_0, L_1)$-smooth for $L_1 = 1$, while a simple function $x^4$ is not globally $L$-smooth for any $L$. Moreover, $(L_0, L_1)$-smoothness also encompasses all functions that belongs to the so-called exponential family. Figure 1(a) presents some simple examples and Figure 1(b) shows that the local smoothness of $(L_0, L_1)$-smooth functions strongly correlates to the gradient norm.

*Secondly*, Zhang et al. [2020a] performed experiments to show that $(L_0, L_1)$-smoothness is a preciser characterization of the landscapes for objective functions in many real-world tasks, especially for training a deep neural network model. It was observed that the local Lipschitz constant $L_0$ near the stationary point is thousands of times smaller than the global one $L$ in the LSTM training (see Figure 1(c) taken from Zhang et al. [2020a]).

Seeing this, it is desirable to give a comprehensive and deep analysis on iteration complexities for $(L_0, L_1)$-smooth objectives. How fast can we achieve to find a first-order stationary point for $(L_0, L_1)$-smooth functions? What are simple algorithms that provably achieve such a convergence rate? In this paper, we give *affirmative* answers to the above questions. In fact, due to the violent fluctuation of gradients, the efficiency of (stochastic) Gradient Descent with a constant step size degenerates, whereas we will show in this paper that by simply combining the clipping technique, a wide range of algorithms can achieve much better convergence rate for $(L_0, L_1)$-smooth functions. In fact, when $\varepsilon$ is small, the complexities (i.e. the number of gradient queries required) are

$\mathcal{O}\left(\Delta L_0 \varepsilon^{-2}\right)$ for the deterministic setting and $\mathcal{O}\left(\Delta L_0 \sigma^2 \varepsilon^{-4}\right)$ for the stochastic setting (see Section 3 for details), which are both independent of $L_1$. Compared with Zhang et al. [2020a] who only studied clipped (stochastic) gradient descent, we consider proposing a unified framework which contains a variety of clipping-based algorithms and achieve much sharper complexities. The main technique for our proof is by introducing a novel Lyapunov function which does not appear in existing studies. We believe that our work provides better understandings for the clipping technique in training deep neural networks. We summarize the contributions of the paper in the following.

- We provide a general framework to analyze the clipping technique for optimizing $(L_0, L_1)$-smooth functions. It contains a variety of clipping algorithms, including gradient clipping and momentum clipping as special cases.

- We provide convergence analysis for the general framework we propose. We show that our bounds are *tight* by comparing with existing lower bounds. For gradient clipping, a special case in our framework, our result is much sharper than that proposed by Zhang et al. [2020a].

- We conduct experiments on a variety of different tasks, and observe that the clipping algorithms consistently perform better than vanilla ones.

**Notations.** For a vector $x \in \mathbb{R}^d$, we denote $\|x\|$ as the $l_2$-norm of $x$. For a matrix $A \in \mathbb{R}^{m \times n}$, let $\|A\|$ be the spectral norm of $A$. Given functions $f, g : \mathcal{X} \to [0, \infty)$ where $\mathcal{X}$ is any set, we say $f = \mathcal{O}(g)$ if there exists a constant $c > 0$ such that $f(x) \leq cg(x)$ for all $x \in \mathcal{X}$, and $f = \Omega(g)$ if there exists a constant $c > 0$ such that $f(x) \geq cg(x)$ for all $x \in \mathcal{X}$. We say $f = \Theta(g)$ if $f = \mathcal{O}(g)$ and $f = \Omega(g)$.

## 1.1 Related Work

**Clipping/normalizing Techniques.** Clipping/normalizing has long been a popular technique in optimizing large-scale non-convex optimization problems (e.g. [Mikolov, 2012, Pascanu et al., 2013, Goodfellow et al., 2016, You et al., 2017]). There are several views which provide understandings for the clipping and normalizing techniques. Some show that clipping can reduce the stochastic noise. For example, Zhang et al. [2019], Gorbunov et al. [2020] showed that clipping is crucial for convergence when the stochastic gradient noise is heavy-tailed. Menon et al. [2020] pointed out that clipping can mitigate the effect of label noise. Cutkosky and Mehta [2020] found that adding momentum in normalized SGD provably reduces the stochastic noise. Another line of works try to understand the function of clipping and normalizing for the standard smooth optimization. For example, Levy [2016] showed that normalized GD can provably escape saddle points. Fang et al. [2018] designed a new algorithm based on normalized GD which achieves a faster convergence rate under suitable conditions. Gradient clipping has also been used to design differentially private optimization algorithms [Abadi et al., 2016].

The work of Zhang et al. [2020a] is mostly related to this paper. A detailed comparison between the two works is shown in Subsection 2.2.

**Lower Bounds For Non-convex Optimization.** A series of recent works establish lower bounds for finding an $\varepsilon$-stationary point of a general non-convex and $L$-smooth function, either in deterministic setting [Carmon et al., 2019] or in stochastic setting [Drori and Shamir, 2019, Arjevani et al., 2019]. In this paper we borrow their counter examples to show the tightness of our obtained complexities for the general $(L_0, L_1)$-smooth functions.

## 2 Assumptions & Comparisons of Results

### 2.1 Assumptions

We first present the assumptions that will be used in our theoretical analysis, which follows from Zhang et al. [2020a].

**Assumption 2.1** *We assume* $\Delta := F(x_0) - F^* < \infty$ *where* $F^* = \inf_{x \in \mathbb{R}^d} F(x)$ *is the global infimum value of* $F(x)$.

**Assumption 2.2** *We assume that* $F(x)$ *is* $(L_0, L_1)$-*smooth.*

**Remark 2.3** *This assumption can be relaxed to the following: there exists $K_0, K_1 > 0$ such that for all $x, y \in \mathbb{R}^d$, if $\|x - y\| \leq \frac{1}{K_1}$, then*

$$\|\nabla F(x) - \nabla F(y)\| \leq (K_0 + K_1 \|\nabla F(y)\|)\|x - y\|$$

*It does not need $F$ to be twice differentiable and is strictly weaker than $L$-smoothness.*

In the stochastic setting, for the briefness of our analysis, we assume that the noise is unbiased and bounded.

**Assumption 2.4** *For all $x \in \mathbb{R}^d$, $\mathbb{E}_\xi [\nabla f(x, \xi)] = \nabla F(x)$. Furthermore, there exists $\sigma > 0$ such that for all $x \in \mathbb{R}^d$, the noise satisfies $\|\nabla f(x, \xi) - \nabla F(x)\| \leq \sigma$ with probability 1.*

Note that another commonly used noise assumption in the optimization literature is *bounded variance* assumption, i.e. $\mathbb{E}_\xi \left[ \|\nabla f(x, \xi) - \nabla F(x)\|^2 \right] \leq \sigma^2$, therefore Assumption 2.4 is stronger than it. However, we adopt Assumption 2.4 as it is also used in the original work of gradient clipping in Zhang et al. [2020a], and it makes our analysis much simpler.

## 2.2 Comparisons of Results

In this paper, we mainly take our concern on the stochastic setting, though we also establish the complexities for the deterministic setting. We summarize the comparison in the stochastic setting with existing complexity results in Table 1, in which we only present the dominating complexities with respect to $\varepsilon$.

Table 1: Comparisons of gradient complexity in the stochastic setting.

| Algorithms | Complexities* |
|---|---|
| SGD [Ghadimi and Lan, 2013] | $\mathcal{O}\left(\Delta(L_0 + L_1 M)\sigma^2 \varepsilon^{-4}\right)$ ** |
| Clipped SGD [Zhang et al., 2020a] | $\mathcal{O}\left((\Delta + (L_0 + L_1\sigma)\sigma^2 + \sigma L_0^2/L_1)^2 \varepsilon^{-4}\right)$ |
| Clipping Framework (this paper) | $\mathcal{O}\left(\Delta L_0 \sigma^2 \varepsilon^{-4}\right)$ |
| Lower Bound | $\Omega\left(\Delta L_0 \sigma^2 \varepsilon^{-4}\right)$ *** |

*For clarity, we only present the dominating term (with respect to $\varepsilon$) here.
**For SGD, we further assume the gradient norm is upper bounded by $M$.
***See section 3.3 for a detailed discussion of the lower bound.

For standard SGD, if we further assume that the gradient is upper bounded by $M$, i.e. $M := \sup_{x \in \mathbb{R}^d} \|\nabla f(x)\| < \infty$, Assumption 2.2 leads to an upper bound of the global Lipschitz constants $L := L_0 + L_1 M$. Therefore, the standard results for $L$-smooth functions (e.g. [Ghadimi and Lan, 2013]) implies that Gradient Descent with a constant step size can achieve complexity of $\mathcal{O}(\Delta(L_0 + L_1 M)\sigma^2 \varepsilon^{-4})$ for finding a first-order stationary point in the stochastic setting[3]. However, the upper bound of the gradient $M$ is typically very large, especially when the parameters have a poor initialization, which makes SGD converges arbitrarily slow. In contrast, our result indicates that the clipping framework (shown in Algorithm 1 which includes a variety of clipping methods) achieves complexity of $\mathcal{O}(\Delta L_0 \sigma^2 \varepsilon^{-4})$, therefore the dominating term of our bound is independent of both $M$ and $L_1$. This provides a strong justification for the efficacy of clipping methods.

Compared with the bound for clipped SGD established in Zhang et al. [2020a], our results improve theirs on the dependencies for all problem-dependent parameters, i.e. $\Delta$, $\sigma$, $L_0$, and especially $L_1$ by order. For SGD with arbitrarily chosen step sizes (thus include clipped SGD), the example in Drori and Shamir [2019] can be used to show that clipped SGD is optimal (cf. Section 3.3).

## 3 General Analysis of Clipping

We aim to present a general framework in which we can provide a unified analysis for commonly used clipping-based algorithms. Since momentum is one of the most popular acceleration technique in optimization community, our framework takes this acceleration procedure into account. We show our framework in Algorithm 1, where we can simply replace $\nabla f(x_t, \xi_t)$ by $\nabla F(x_t)$ for the deterministic setting.

**Algorithm 1:** The General Clipping Framework

---

**Input :** Initial point $x_0$, learning rate $\eta$, clipping parameter $\gamma$, momentum $\beta \in [0, 1)$,
 interpolation parameter $\nu \in [0, 1]$ and the total number of iterations $T$

1  Initialize $m_0$ arbitrarily;
2  **for** $t \leftarrow 0$ *to* $T - 1$ **do**
3  $\quad$ Compute the stochastic gradient $\nabla f(x_t, \xi_t)$ for the current point $x_t$;
4  $\quad$ $m_{t+1} \leftarrow \beta m_t + (1 - \beta) \nabla f(x_t, \xi_t)$;
5  $\quad$ $x_{t+1} \leftarrow x_t - \left[ \nu \min\left( \eta, \dfrac{\gamma}{\|m_{t+1}\|} \right) m_{t+1} + (1 - \nu) \min\left( \eta, \dfrac{\gamma}{\|\nabla f(x_t, \xi_t)\|} \right) \nabla f(x_t, \xi_t) \right]$;

---

We notice that our framework is similar to the Quasi-Hyperbolic Momentum(QHM) algorithm proposed by Ma and Yarats [2018], while they did not consider the clipping technique. They pointed out that QHM contains a wide range of popular algorithms (e.g. SGD+momentum, Nesterov Accelerated SGD, AccSGD, etc). As a result, for different choice of hyper-parameters, our framework encompasses the clipping version of all these algorithms. We now discuss several representative examples in our framework.

- **Gradient Clipping.** By choosing $\nu = 0$ in Algorithm 1, we obtain the *clipped GD/SGD algorithm* which can be written as:

$$x_{t+1} \leftarrow x_t - \min\left( \eta, \gamma/\|\nabla f(x_t, \xi_t)\| \right) \nabla f(x_t, \xi_t)$$

It follows that in gradient clipping, the gradient is clipped to have its norm no more than $\gamma/\eta$.

- **Momentum Clipping.** By choosing $\nu = 1$ in Algorithm 1, we perform the update using a clipped version of momentum which can be written as:

$$x_{t+1} \leftarrow x_t - \min\left( \eta, \gamma/\|m_{t+1}\| \right) m_{t+1}$$

The approach has already been used in previous works [Zhang et al., 2019, 2020b], albeit in different settings. To the best of our knowledge, there is no existing analysis of this algorithm even for optimizing standard $L$-smooth functions.

- **Mixed Clipping.** By choosing $\nu \in (0, 1)$, we obtain the mixed clipping algorithm. Although this form of clipping is not widely used in practice, we observe from experiments that it typically converges faster than both gradient clipping and momentum clipping. Some explanations of this observation are provided in Appendix E.

- **Normalized Momentum.** By choosing $\nu = 1$ and $\eta \to +\infty$ in Algorithm 1, we recover the *normalized SGD+momentum algorithm*. This algorithm performs a normalized (rather than clipped) update in each iteration. It has been analyzed in Cutkosky and Mehta [2020] for $L$-smooth functions and a layer-wise variant was used in the LARS algorithm [You et al., 2017]. We will provide a detailed discussion of this algorithm in the Appendix C.

### 3.1 Main Results

In this section we first deal with the deterministic case, in which we can get a strong justification that clipping is a natural choice to optimize $(L_0, L_1)$-smooth functions. We have the following Theorem.

**Theorem 3.1** *[Convergence of Algorithm 1, Deterministic Setting] Let the function $F$ satisfy Assumptions 2.1 and 2.2. Set $m_0 = \nabla F(x_0)$ in Algorithm 1 for simplicity. Fix $\varepsilon > 0$ be a small constant. For any $0 \le \beta < 1$ and $0 \le \nu \le 1$, if $\gamma \le \frac{1-\beta}{10BL_1}$ and $\eta \le \frac{1-\beta}{10AL_0}$ where $A = 1.06$, $B = 1.06$, then*

$$\frac{1}{T} \sum_{t=1}^{T} \|\nabla F(x_t)\| \le 2\varepsilon$$

*as long as*

$$T \ge 3\Delta \max\left\{ \frac{1}{\varepsilon^2 \eta}, \frac{25\eta}{\gamma^2} \right\}. \tag{2}$$

In Theorem 3.1, the $(L_0, L_1)$-smoothness is precisely reflected in the restriction of hyper-parameters $\gamma = \mathcal{O}(1/L_1)$ and $\eta = \mathcal{O}(1/L_0)$. For large $L_1$, we must use a small clipping hyper-parameter to guarantee convergence. This also coincides with the intuition that in highly non-smooth regions we should take a small step.

Theorem 3.1 states that in the deterministic setting, for any $\varepsilon > 0$, our framework can find an $\varepsilon$-approximate stationary point in $\mathcal{O}\left(\Delta \max\left\{\frac{L_0}{\varepsilon^2}, \frac{L_1^2}{L_0}\right\}\right)$ gradient evaluations if we choose $\gamma = \Theta(1/L_1)$ and $\eta = \Theta(1/L_0)$. When $\varepsilon = \mathcal{O}(L_0/L_1)$, the dominating term is $\mathcal{O}\left(\Delta L_0 \varepsilon^{-2}\right)$.

Now we turn to our main result in the stochastic setting. We have the following theorem.

**Theorem 3.2** *[Convergence of Algorithm 1, Stochastic setting] Let the function $F$ satisfy Assumptions 2.1 and 2.2, and the noise satisfies Assumption 2.4 with $\sigma \geq 1$. Set $m_0 = \nabla F(x_0)$ in Algorithm 1 for simplicity. Fix $0 < \varepsilon \leq 0.1$ be a small constant. For any $0 \leq \beta < 1$ and $0 \leq \nu \leq 1$, if $\gamma \leq \frac{\varepsilon}{2\sigma} \min\left\{\frac{\varepsilon}{AL_0}, \frac{1-\beta}{AL_0}, \frac{1-\beta}{25BL_1}\right\}$ and $\gamma/\eta = 5\sigma$ where constants $A = 1.01, B = 1.01$, then*

$$\frac{1}{T} \sum_{t=1}^{T} \mathbb{E}\|\nabla F(x_t)\| \leq 3\varepsilon \tag{3}$$

*as long as*

$$T \geq \frac{3}{\varepsilon^2 \eta}\Delta. \tag{4}$$

*Here the expectation is taken over all the randomness $\xi_0, \cdots, \xi_{T-1}$.*

Theorem 3.2 shows that in the stochastic setting, for any $\varepsilon > 0$, our framework can find an $\varepsilon$-approximate stationary point in $\mathcal{O}\left(\Delta\sigma^2\left(\max\left\{\frac{L_0}{\varepsilon^4}, \frac{L_1^4}{L_0^3}\right\}\right)\right)$ gradient evaluations. When $\varepsilon < \min\left\{1, \frac{L_0}{25L_1}\right\}(1-\beta)$, the term $\min\left\{\frac{\varepsilon}{AL_0}, \frac{1-\beta}{AL_0}, \frac{1-\beta}{25BL_1}\right\}$ reduces to $\frac{\varepsilon}{AL_0}$. In this case $L_1$ no longer affects the choice of steps sizes $\eta$ and $\gamma$, and the complexity in (4) reduces to $\mathcal{O}\left(\Delta L_0 \sigma^2 \varepsilon^{-4}\right)$.

Theorem 3.2 suggests that the clipping threshold should take $\gamma/\eta = \Theta(\sigma)$, which only depends on the noise and is several times larger than the its variance. This matches previous understanding of gradient clipping, in that clipping the stochastic gradient controls the variance while introducing some additional bias, and the clipping threshold should be tuned to trade-off variance with the introduced bias [Zhang et al., 2019].

We emphasize that in both settings, the dominating terms in our upper bounds are independent of the gradient upper bound $M$ and the smoothness parameter $L_1$. In other words, the efficiency of Algorithm 1 is essentially unaffected by these quantities. Recall that $M$ and $L_1$ are related to steep cliffs in the landscape where the gradient may be large or fluctuate violently. Therefore, our results suggest that such non-smoothness can be tackled with clipping methods without sacrificing efficiency.

## 3.2 Proof Sketch

The analysis of Algorithm 1 is in fact challenging, as it uses both momentum and adaptive step sizes. Also, the general $(L_0, L_1)$-smoothness assumption makes things more complicated. In this subsection we briefly introduce our proof technique. We hope our proof is also useful to a better understanding of other adaptive algorithms that combine momentum (such as Adam [Kingma and Ba, 2014]).

**Proof sketch of Theorem 3.1.** Due to the momentum term, each step in Algorithm 1 is not necessarily a descent one, which makes it difficult to prove convergence using traditional techniques. Instead, we construct a novel *Lyapunov function* as follows:

$$G(x, m) = F(x) + \frac{\beta\nu}{2(1-\beta)} \min\left(\eta\|m\|^2, \gamma\|m\|\right) \tag{5}$$

We aim to analyze the descent property of the sequence $\{G(x_t, m_t)\}_{t=0}^{T}$. Define $\rho = \gamma/\eta$, $\mathcal{S} := \{t \in \mathbb{N} : t < T, \max(\|\nabla F(x_t)\|, \|m_t\|, \|m_{t+1}\|) \geq \rho\}$ and $\overline{\mathcal{S}} = \{t \in \mathbb{N} : t < T\}\backslash\mathcal{S}$. Let

$T_{\mathcal{S}} = |\mathcal{S}|$. We separately provide one-step analysis for the two cases, as stated in Lemma 3.3 and 3.4 respectively.

**Lemma 3.3** *For any $t \in \mathcal{S}$, we have*

$$G(x_t, m_t) - G(x_{t+1}, m_{t+1}) = \Omega\left(\gamma(1-\gamma)(\|\nabla F(x_t)\| + \rho) - \gamma\|\nabla F(x_t) - m_t\|\right) \quad (6)$$

We prove this lemma by using the $(L_0, L_1)$-smoothness properties deduced in Appendix A and conducting a comprehensive discussion on three cases in Lemmas B.1-B.3. Furthermore, we show in Lemma B.4 that $\sum_{t \in \mathcal{S}} \|\nabla F(x_t) - m_t\| = \mathcal{O}\left(\gamma T_{\mathcal{S}}(\rho + \sum_{t \in \mathcal{S}} \|\nabla F(x_t)\|)\right)$. By choosing a small enough $\gamma$ and carefully dealing with constants, we can conclude that the total amount of decrease of the Lyapunov function is $\Omega\left(\rho\gamma T_{\mathcal{S}}\right)$.

**Lemma 3.4** *For any $t \in \overline{\mathcal{S}}$, if $\eta = \mathcal{O}(1/L_0)$ and $\gamma = \mathcal{O}(1/L_1)$, then we have*

$$G(x_t, m_t) - G(x_{t+1}, m_{t+1}) = \Omega\left(\eta\left((1 - \nu\beta)\|\nabla F(x_t)\|^2 + \nu\beta\|m_t\|^2\right)\right) \quad (7)$$

We prove (7) in Lemma B.5-B.7 by the fact that Algorithm 1 performs an unclipped update if $t \in \overline{\mathcal{S}}$. Since the bound (7) is small in term of $\|\nabla F(x_t)\|$ if $\beta$ and $\nu$ are close to 1, we convert $\|m_t\|$ to $\|\nabla F(x_t)\|$ by proving that $\sum_{t \in \overline{\mathcal{S}}} \|m_t\| = \Omega\left(\sum_{t \in \overline{\mathcal{S}}} \|\nabla F(x_t)\| - \gamma(\rho T_{\mathcal{S}} + \sum_{t \in \mathcal{S}} \|\nabla F(x_t)\|)\right)$ in Lemma B.8. The term related to $\mathcal{S}$ can all be offset by the terms in Lemma 3.3, and the term related to $\overline{\mathcal{S}}$ combined with $\eta(1 - \nu\beta)\|\nabla F(x_t)\|^2$ can be shown to ensure a descent amount of $\Omega\left(\varepsilon\eta\|\nabla F(x_t)\| - \varepsilon^2\eta\right)$, which is $\Omega\left(\varepsilon^2\eta\right)$ as long as $\|\nabla F(x_t)\| \geq 2\varepsilon$.

Finally, by combining the above two cases we obtain the conclusion in Theorem 3.1.

**Proof sketch of Theorem 3.2.** In the stochastic setting, it requires a different treatment to deal with the noise. We define the *true momentum* $\tilde{m}$ recursively by

$$\tilde{m}_{t+1} = \beta\tilde{m}_t + (1 - \beta)\nabla F(x_t) \quad (8)$$

where $\tilde{m}_0 = m_0$, and analyze the descent property of the following sequence $\{G(x_t, \tilde{m}_t)\}_{t=0}^T$. We also consider two cases: $\max(5\|\nabla F(x_t)\|/4, \|\tilde{m}_t\|, \|m_{t+1}\|) \geq \rho$ and $\max(5\|\nabla F(x_t)\|/4, \|\tilde{m}_t\|, \|m_{t+1}\|) < \rho$. We split $m_t$ into $\tilde{m}_t$ and $m_t - \tilde{m}_t$ such that the latter term is merely composed of noises $\nabla f(x_\tau, \xi_\tau) - \nabla F(x_\tau)$ $(\tau < t)$. While most of the procedure (Lemmas B.10-B.14) parallels the deterministic setting, there are two additional challenges due to the presence of noise:

- *Firstly*, since the gradients are not exact, the stochastic gradient we have access to is not guaranteed to be small even for the case of $t \in \overline{\mathcal{S}}$. Fortunately, the choice of parameters in Theorem 3.2 ($\rho = 5\sigma$) settles such difficulty.

- *Secondly*, we need to deal with the noise in momentum, i.e. $m_t - \tilde{m}_t$. In particular, we use a recursive argument to obtain a good bound of $\mathbb{E}\langle\nabla F(x_t), m_{t+1} - \tilde{m}_{t+1}\rangle$.

Finally, by choosing proper $\eta$ and $\gamma$, we can obtain Theorem 3.2.

### 3.3 Lower Bounds and Discussions

Theorem 3.1 and 3.2 provide *upper bounds* for the complexity of Algorithm 1. Now we compare these results with existing lower bounds and discuss the tightness of our results.

**Deterministic Setting.** Carmon et al. [2019] have shown that there exists an $L$-smooth function $F$ such that any (possibly randomized) algorithm requires at least $\Omega\left(\Delta L\varepsilon^{-2}\right)$ queries to gradient to ensure finding a point $x$ such that $\|\nabla F(x)\| \leq \varepsilon$. Since Assumption 2.2 is weaker than $L$-smoothness ($L_0 \leq L$), we have that the lower bound for $(L_0, L_1)$-smooth functions is $\Omega\left(\Delta L_0\varepsilon^{-2}\right)$. From Theorem 3.1, Algorithm 1 is *optimal* since it can achieve the lower bound when ignoring numerical constants.

**Stochastic Setting.** From the example constructed in Drori and Shamir [2019], we have that for any *SGD* method with arbitrary (possibly adaptive) step sizes and aggregation schemes[4], the complexity

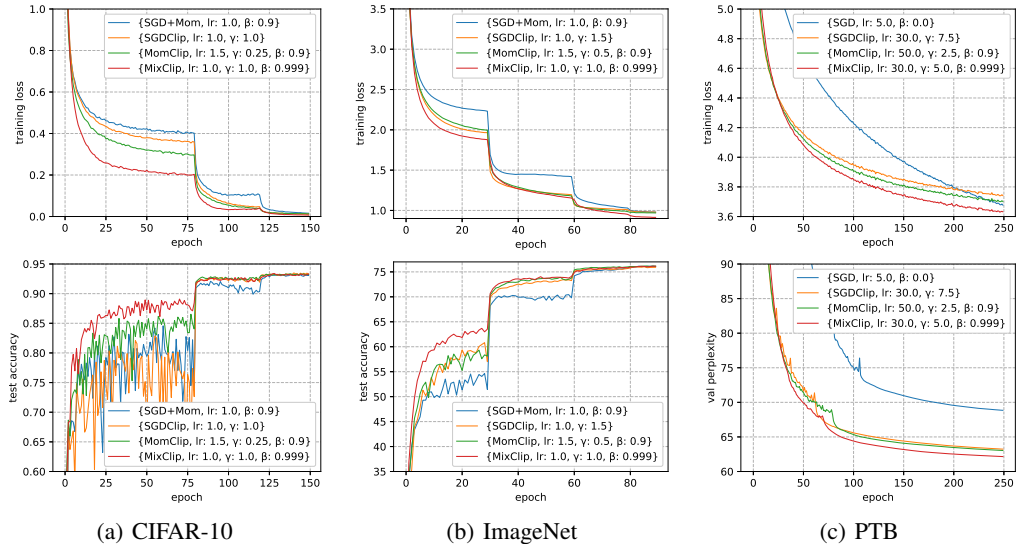

Figure 2: Training loss curve and test accuracy/perplexity curve on CIFAR-10, ImageNet and PTB datasets. All clipping algorithms outperform SGD. Mixed clipping has the best training speed on these three datasets.

lower bound is exactly $\Omega\left(\Delta L_0 \sigma^2 \varepsilon^{-4}\right)$ for $(L_0, L_1)$-smooth functions that have $\sigma$-bounded gradient noises. Therefore Theorem 3.2 indicates that clipped SGD *matches* the lower bound.

One may ask : what is the lower bound for general stochastic gradient-based algorithms? In fact, from the example in Arjevani et al. [2019], we have that any algorithm needs $\Omega\left(\Delta L_0 \sigma_1^2 \varepsilon^{-4}\right)$ stochastic gradient queries to find an $\varepsilon$-approximate stationary point for a hard $(L_0, L_1)$-smooth function whose gradient noise has a $\sigma_1^2$-bounded variance. It is our conjecture that the lower bound for optimizing $(L_0, L_1)$-smooth functions that have $\sigma$-bounded gradient noises is also $\mathcal{O}\left(\Delta L_0 \sigma^2 \varepsilon^{-4}\right)$. We leave the study as a future work.

# 4   Experiments

We conduct extensive experiments and find the clipping algorithms indeed consistently outperform their unclipped counterpart. We present experimental results on three deep learning benchmarks: CIFAR-10 classification using ResNet-32, Imagenet classification using ResNet-50 and language modeling on Penn Treebank (PTB) dataset using AWD-LSTM. We put all the experimental details in the Appendix. Our code is available at `https://github.com/zbh2047/clipping-algorithms`.

**CIFAR-10 classification with ResNet.** We train the standard ResNet-32 [He et al., 2016] architecture on CIFAR-10. We use SGD with momentum for the baseline algorithm with a decaying learning rate schedule, which is the standard choice to train the ResNet architecture. We set learning rate $\eta = 1.0$, momentum $\beta = 0.9$ and minibatch size 128, following the common practice. For all the clipping algorithms, we choose the best $\eta$ and $\gamma$ based on a course grid search, while keeping other hyper-parameters and training strategy the same as SGD+momentum. We simply set the hyper-parameters $\nu = 0.7$ and $\beta = 0.999$ in mixed clipping, as suggested in Ma and Yarats [2018] (for its unclipped counterpart QHM). We run 5 times for each algorithm using different random seeds to make the results more reliable.

Figures 2(a) demonstrates the results. It can be seen that all the algorithms achieve a test accuracy more than 93% on CIFAR-10. Note that all clipping algorithms converge faster than SGD+momentum. Particularly, the mixed clipping (Algorithm 1) outperforms SGD+momentum by a large margin in term of training speed. As a result, one can possibly adopt a more aggressive learning rate decaying schedule to reduce training time considerably.

**ImagNet classification with ResNet.** We train the standard ResNet-50 [He et al., 2016] architecture on ImageNet. For the baseline algorithm, we choose SGD with learning rate $lr = 1.0$ and momentum $\beta = 0.9$, following Goyal et al. [2017]. We use batch size 256 on 4 GPUs.

Figure 2(b) plot the training loss curve and validation accuracy curve on ImageNet. All the algorithms reach a validation accuracy of about 76%. However, all the clipping algorithms train faster than the baseline SGD. Mixed clipping performs the best among the four algorithms.

**Language modeling with LSTM.** We train the state-of-the-art AWD-LSTM [Merity et al., 2017] on Penn Treebank (PTB) dataset [Mikolov et al., 2010]. We first follow the training strategy in Merity et al. [2017], where they use averaged SGD without momentum with learning rate $\eta = 30$ and clipping parameter $\gamma = 7.5$. Since our purpose is to compare different algorithms rather than to achieve state-of-the-art results, we only train AWD-LSTM for 250 epochs. We then evaluate other algorithms including standard SGD without clipping, momentum clipping, and mixed clipping. We choose the best $\eta$ and $\gamma$ (using validation perplexity criterion) based on a course grid search. Results are shown in Figure 2(c).

Figure 2(c) clearly shows all clipping methods converge much faster than SGD without clipping, and are much better in term of validation perplexity. This is consistent with our theory, in that the vanilla SGD must use a very small learning rate to guarantee convergence [Zhang et al., 2020a], which will be slow and be harmful to generalization on validation set according to previous works [Huang et al., 2017, Kleinberg et al., 2018] . Therefore clipping technique is crucial in LSTM models. We can also find that the training and test curve of mixed clipping is much better than both gradient clipping and momentum clipping. The mixed clipping improves validation perplexity for more than 1 point compared to clipped SGD after 250 epochs.

## 5    Conclusion

This paper proposes a detailed study for clipping methods under a general framework. In particular, we explore the possibility of combining clipping with other popular techniques, e.g. momentum acceleration, in deep learning. We provide a general and tight analysis for the framework, showing the efficiency of clipping methods in optimizing a class of non-convex and non-smooth (in traditional sense) functions. Experiments confirm that these methods have superior performance. We hope that our work affords more understandings on the clipping technique and $(L_0, L_1)$ smooth functions.

There are still many open questions that have not yet been answered. Firstly, as discussed in Section 3.3, we are not aware of any lower bounds for general first-order methods that can be applied our setting. Thus, it is interesting to explore such lower bound, or to relax Assumption 2.4 to the more general bounded variance assumption. Secondly, although we have shown the superiority of clipping-based methods, we do not provide theoretical explanation why some clipping schemes are better than others as observed in experiments. We believe that this can only be done by exploring new and better smoothness assumptions. Thirdly, the empirical superiority of other adaptive methods ( e.g. AdaGrad [Duchi et al., 2011], Adam [Kingma and Ba, 2014] ) have not been justified from a theoretical point of view. We hope that our analysis is helpful for the analysis of these methods. Finally, we are looking forward to seeing better optimization algorithms with better convergence properties in future work.

## Broader Impact

Deep neural networks have achieved great success in recent years. In this paper, we provide a strong justification for the clipping technique in training deep neural networks and provides a satisfactory answer on how to efficiently optimize a general possibly non-convex $(L_0, L_1)$-smooth objective function. It closely aligns with the community's pursuit of explainability, controllability, and practicability of machine learning.

Besides its efficiency in training deep neural networks, a series of recent work ( Thakkar et al. [2019], Chen et al. [2020], Lee and Kifer [2020] ) also studies the relation between clipping and privacy preservation, which appears to be a major concern in machine learning applications. Therefore, we hope that a thorough understanding of clipping methods will be beneficial to the modern society.

## Acknowledgement

This work was supported by National Key R&D Program of China (2018YFB1402600), Key-Area Research and Development Program of Guangdong Province (No. 2019B121204008)] and Beijing Academy of Artificial Intelligence.

## Footnotes

[3]We will show in Appendix D that even using Assumption 2.4, such upper bound can not be improved.

[4]See details in Appendix D.

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
