[Supplementary Material]

# Supplementary Material for *Improved Analysis of Clipping Algorithms for Non-convex Optimization*

## 1 Appendix A   Properties of $(L_0, L_1)$-smooth functions

2  In this section, we prove some important properties of $(L_0, L_1)$-smooth functions. These properties
3  will be frequently used in subsequent sections.

4  We first present a basic lemma without proof.

5  **Lemma A.1** *(Grönwall's inequality) [Gronwall, 1919] Let $I = [a, b]$ denote an interval of the real*
6  *line with $a < b$. Let $f, g, h$ be continuous real-valued functions defined on $I$. Assume $g$ is non-*
7  *decreasing, $h$ is non-negative, and the negative part of $g$ is integrable on every closed and bounded*
8  *subinterval of $I$.If*

$$f(t) \le g(t) + \int_a^t h(s)f(s)\mathrm{d}s, \quad \forall t \in I, \tag{1}$$

9  *then*

$$f(t) \le g(t) \exp\left(\int_a^t h(s)\mathrm{d}s\right), \quad \forall t \in I. \tag{2}$$

10  The following result, Lemma A.2 , is a generalization of Lemma 9 in Zhang et al. [2020].

11  **Lemma A.2** *Let $F$ be $(L_0, L_1)$-smooth, and $c > 0$ be a constant. Given $x$, for any $x^+$ such that*
12  $\|x^+ - x\| \le c/L_1$, *we have* $\|\nabla f(x^+)\| \le e^c \left(\frac{cL_0}{L_1} + \|\nabla F(x)\|\right)$.

**Proof:**  Let $\gamma(t)$ be defined as $\gamma(t) = t(x^+ - x) + x, t \in [0, 1]$, then we have

$$\nabla F(\gamma(t)) = \int_0^t \nabla^2 F(\gamma(\tau))\left(x^+ - x\right)\mathrm{d}\tau + \nabla F(\gamma(0))$$

13  We then bound the norm of $\nabla F(\gamma(t))$:

$$\|\nabla F(\gamma(t))\| \le \int_0^t \|\nabla^2 F(\gamma(\tau))\left(x^+ - x\right)\|\mathrm{d}\tau + \|\nabla F(\gamma(0))\| \tag{3}$$

$$\le \|x^+ - x\| \int_0^t \|\nabla^2 F(\gamma(\tau))\|\,\mathrm{d}\tau + \|\nabla F(x)\| \tag{4}$$

$$\le \frac{c}{L_1} \int_0^t \left(L_0 + L_1\|\nabla F(\gamma(\tau))\|\right)\mathrm{d}\tau + \|\nabla F(x)\| \tag{5}$$

14  The first inequality uses the triangular inequality of 2-norm; The second inequality uses the property
15  of spectral norm; The third inequality uses the definition of $(L_0, L_1)$-smoothness. By applying the
16  Grönwalls inequality we get

$$\|\nabla F(\gamma(t))\| \le \left(\frac{L_0}{L_1}ct + \|\nabla F(x)\|\right)\exp(ct) \tag{6}$$

17  The Lemma follows by setting $t = 1$. □

18  Now we are able to prove a *descent inequality*, which is similar to the descent inequality for $L$-
19  smooth functions. In fact, if a function $F$ is $L$-smooth, it is well-known that for any $x, y$, we have
20

$$F(y) \le F(x) + \langle \nabla F(x), y - x \rangle + \frac{L}{2} \|y - x\|^2$$

22  **Lemma A.3** *(Descent Inequality) Let $F$ be $(L_0, L_1)$-smooth, and $c > 0$ be a constant. For any $x_k$*
23  *and $x_{k+1}$, as long as $\|x_k - x_{k+1}\| \le c/L_1$, we have*

$$F(x_{k+1}) \le F(x_k) + \langle \nabla F(x_k), x_{k+1} - x_k \rangle + \frac{AL_0 + BL_1 \|\nabla F(x_k)\|}{2} \|x_{k+1} - x_k\|^2 \quad (7)$$

24  *where $A = 1 + e^c - \frac{e^c - 1}{c}, B = \frac{e^c - 1}{c}$.*

25  **Proof:**  Let $\gamma(t)$ be defined as $\gamma(t) = t(x_{k+1} - x_k) + x_k, t \in [0, 1]$. The following derivation
26  uses Taylor's theorem (in (8)), then uses triangular inequality, Cauchy-Schwarz inequality and the
27  property of spectral norm (in (9)):

$$F(x_{k+1}) \le F(x_k) + \langle \nabla F(x_k), x_{k+1} - x_k \rangle + \int_0^1 (x_{k+1} - x_k)^T \nabla^2 F(\gamma(t))(x_{k+1} - \gamma(t)) \mathrm{d}t$$

$$(8)$$

$$\le F(x_k) + \langle \nabla F(x_k), x_{k+1} - x_k \rangle + \int_0^1 \|(x_{k+1} - x_k)\| \|\nabla^2 F(\gamma(t))\| \|x_{k+1} - \gamma(t)\| \mathrm{d}t$$

$$(9)$$

$$= F(x_k) + \langle \nabla F(x_k), x_{k+1} - x_k \rangle + \frac{\|x_{k+1} - x_k\|^2}{2} \int_0^1 \|\nabla^2 F(\gamma(t))\| \mathrm{d}t \quad (10)$$

28  Then we use $(L_0, L_1)$-smoothness and (6) to bound $\|\nabla^2 F(\gamma(t))\|$:

$$\|\nabla^2 F(\gamma(t))\| \le L_0 + L_1 \|\nabla F(\gamma(t))\|$$
$$\le L_0 + L_1 \left( \frac{L_0}{L_1} ct + \|\nabla F(x_k)\| \right) \exp(ct) \quad (11)$$

29  Taking integration we get

$$\int_0^1 \|\nabla^2 F(\gamma(t))\| \mathrm{d}t \le L_0 \left( 1 + e^c - \frac{e^c - 1}{c} \right) + \frac{e^c - 1}{c} L_1 \|\nabla F(x_k)\| \quad (12)$$

30  Substituting (12) into (10) concludes the proof. □

31  **Corollary A.4** *Let $F$ be $(L_0, L_1)$-smooth, and $c > 0$ be a constant. For any $x_k$ and $x_{k+1}$, as long*
32  *as $\|x_k - x_{k+1}\| \le c/L_1$, we have*

$$\|\nabla F(x_{k+1}) - \nabla F(x_k)\| \le (AL_0 + BL_1 \|\nabla F(x_k)\|) \|x_{k+1} - x_k\| \quad (13)$$

33  *where $A = 1 + e^c - \frac{e^c - 1}{c}, B = \frac{e^c - 1}{c}$.*

**Proof:**

$$\|\nabla F(x_k) - \nabla F(x_k - 1)\| = \left\| \int_0^1 \nabla^2 F(tx_{k-1} + (1 - t)x_k)(x_k - x_{k-1}) \mathrm{d}t \right\|$$
$$\le \int_0^1 \|\nabla^2 F(tx_{k-1} + (1 - tx_k)\| \|x_k - x_{k-1}\| \mathrm{d}t$$

$$(14)$$

34  Using (12) leads to the results. □

35  Finally we prove a result which provides a way to upper-bound the gradient norm. A similar result
36  for $L$-smooth functions is the following: if $F$ is $L$-smooth, then for any $x$, we have

$$\|\nabla F(x)\|^2 \le 2L \left( F(x) - \inf_{y \in \mathbb{R}^d} F(y) \right)$$

**Lemma A.5** *(Bounding the gradient norm) Let $F(x)$ be an $(L_0, L_1)$-smooth function, and $F^*$ be the optimal value. Then for any $x_0$, we have*

$$\min\left(\frac{\|\nabla F(x_0)\|}{L_1}, \frac{\|\nabla F(x_0)\|^2}{L_0}\right) \leq 8(F(x_0) - F^*) \tag{15}$$

**Proof:** Define the constant $c = \frac{L_1\|\nabla F(x_0)\|}{AL_0 + BL_1\|\nabla F(x_0)\|}$ and $A = 1 + e^c - \frac{e^c - 1}{c}, B = \frac{e^c - 1}{c}$. It is easy to see that such $0 \leq c < 1$ exists. Let $\lambda = \frac{1}{AL_0 + BL_1\|\nabla F(x_0)\|}$ and $x = x_0 - \lambda\nabla F(x_0)$. Then $\|x - x_0\| \leq c/L_1$. By the descent inequality we have

$$F^* \leq F(x) \leq F(x_0) - \lambda\|\nabla F(x_0)\|^2 + \frac{AL_0 + BL_1\|\nabla F(x_0)\|}{2}\lambda^2\|\nabla F(x_0)\|^2$$
$$= F(x_0) - \frac{1}{2}\lambda\|\nabla F(x_0)\|^2 \tag{16}$$

If $\|\nabla F(x)\| \geq \frac{AL_0}{BL_1}$, then

$$F(x_0) - F^* \geq \frac{\|\nabla F(x_0)\|}{2\left(\frac{AL_0}{\|\nabla F(x_0)\|} + BL_1\right)} \geq \frac{\|\nabla F(x_0)\|}{4BL_1} \geq \frac{\|\nabla F(x_0)\|}{8L_1} \tag{17}$$

If $\|\nabla F(x)\| < \frac{AL_0}{BL_1}$, then

$$F(x_0) - F^* \geq \frac{1}{2}\lambda\|\nabla F(x_0)\|^2 \geq \frac{\|\nabla F(x_0)\|^2}{4AL_0} \geq \frac{\|\nabla F(x_0)\|^2}{8L_0} \tag{18}$$

$\square$

## A.1 Relaxation of $(L_0, L_1)$-smoothness (Remark 2.3)

The original definition of $(L_0, L_1)$-smoothness requires the function to be twice-differentiable. Under this definition, $(L_0, L_1)$-smoothness is actually *not* weaker than $L$-smoothness, which only requires the function to be continuous differentiable. In this section we prove that the alternative definition provided in Remark 2.3 is sufficient for all the results in this paper.

Now, suppose that there exists $K_0, K_1 > 0$ such that for all $x, y \in \mathbb{R}^d$, if $\|x - y\| \leq \frac{1}{K_1}$, then

$$\|\nabla F(x) - \nabla F(y)\| \leq (K_0 + K_1\|\nabla F(y)\|)\|x - y\| \tag{19}$$

We check that Lemma A.2 and A.3 still holds under the new assumption (with $L_0, L_1$ replaced by $K_0, K_1$, up to numerical constants) We immediately obtain from (19) above that

$$\|\nabla F(x)\| \leq 2\|\nabla F(y)\| + \frac{K_0}{K_1} \tag{20}$$

which is of the same form as Lemma A.2. Next, we have

$$F(y) - F(x) - \langle y - x, \nabla F(x)\rangle$$
$$= \int_0^1 \langle \nabla F(\theta y + (1 - \theta)x) - \nabla F(x), x - y\rangle \, \mathrm{d}\theta$$
$$\leq \int_0^1 \left(K_0\theta\|x - y\|^2 + K_1\theta\|x - y\|^2\|\nabla F(x)\|\right) \mathrm{d}\theta \tag{21}$$
$$\leq \frac{K_0 + K_1\|\nabla F(x)\|}{2}\|x - y\|^2$$

which is of the same form as Lemma A.3.

Since all the other results are established on the basis of these two lemmas, we can see that the conclusion still holds under (19).

## Appendix B   Proof of Theorems

We first prove the deterministic case (Theorem 3.1), then generalize the result to stochastic case (Theorem 3.2). In deterministic case we can use fewer notations, which will make the proof more readable and elegant. The proof in stochastic case will rely on all the techniques used in the deterministic case, as well as some new methods.

### B.1   Proof of Theorem 3.1

To simplify the notation, we write the update formula as

$$
\begin{aligned}
m^+ &= \beta m + (1-\beta)\nabla F(x) \\
x^+ &= x - \left( \nu \min\left(\eta, \frac{\gamma}{\|m^+\|}\right) m^+ + (1-\nu)\min\left(\eta, \frac{\gamma}{\|\nabla F(x)\|}\right)\nabla F(x)\right)
\end{aligned}
\tag{22}
$$

when analyzing a single iteration. The error between $m^+$ and $\nabla F(x)$ is denoted as $\delta = m^+ - \nabla F(x)$. Suppose $\gamma \leq c/L_1$ for some constant $c$, and we denote $A = 1 + e^c - \frac{e^c-1}{c}$ and $B = \frac{e^c-1}{c}$, just the same as in the descent inequality (Lemma A.3).

**Lemma B.1** *Let $\mu \geq 0$ be a real constant. For any vector $u$ and $v$,*

$$
-\frac{\langle u, v\rangle}{\|v\|} \leq -\mu\|u\| - (1-\mu)\|v\| + (1+\mu)\|v-u\|
\tag{23}
$$

**Proof:**

$$
\begin{aligned}
-\frac{\langle u, v\rangle}{\|v\|} &= -\|v\| + \frac{\langle v-u, v\rangle}{\|v\|} \\
&\leq -\|v\| + \|v-u\| \\
&\leq -\|v\| + \|v-u\| + \mu(\|v-u\| + \|v\| - \|u\|) \\
&= -\mu\|u\| - (1-\mu)\|v\| + (1+\mu)\|v-u\|
\end{aligned}
$$

$\square$

To prove the theorem, we will construct an *energy function* and explore the decreasing property of this function. We define the energy function $G(x, m)$ to be

$$
G(x, m) = F(x) + \frac{\nu\beta}{2(1-\beta)}\min\left(\eta\|m\|^2, \gamma\|m\|\right)
\tag{24}
$$

and analyze $G(x^+, m^+) - G(x, m)$. We first bound $\min\left(\eta\|m^+\|^2, \gamma\|m^+\|\right) - \min\left(\eta\|m\|^2, \gamma\|m\|\right)$.

**Lemma B.2** *For any momentum vectors $m$ and $m^+ = \beta m + (1-\beta)\nabla F(x)$, let $\delta = m^+ - \nabla F(x)$, then*

$$
\min\left(\eta\|m^+\|^2, \gamma\|m^+\|\right) - \min\left(\eta\|m\|^2, \gamma\|m\|\right) \leq \frac{2(1-\beta)}{\beta}\gamma\|\delta\|
\tag{25}
$$

**Proof:**   Consider the following three cases:

- $\|m\| \geq \gamma/\eta$. In this case

$$
\begin{aligned}
\min\left(\eta\|m^+\|^2, \gamma\|m^+\|\right) - \min\left(\eta\|m\|^2, \gamma\|m\|\right) &\leq \gamma\|m^+\| - \gamma\|m\| \\
&\leq \gamma\|m^+ - m\| \\
&= \frac{1-\beta}{\beta}\gamma\|\delta\|
\end{aligned}
$$

- $\|m\| < \gamma/\eta$ and $\|m^+\| < \gamma/\eta$. In this case

$$
\begin{aligned}
\min\left(\eta\|m^+\|^2, \gamma\|m^+\|\right) - \min\left(\eta\|m\|^2, \gamma\|m\|\right) &= \eta\|m^+\|^2 - \eta\|m\|^2 \\
&= \eta(\|m^+\| - \|m\|)(\|m^+\| + \|m\|) \\
&\leq \frac{2(1-\beta)}{\beta}\gamma\|\delta\|
\end{aligned}
$$

78 • $\|m\| < \gamma/\eta$ and $\|m^+\| > \gamma/\eta$. In this case

$$\min\left(\eta\|m^+\|^2, \gamma\|m^+\|\right) - \min\left(\eta\|m\|^2, \gamma\|m\|\right) = \gamma\|m^+\| - \eta\|m\|^2$$

$$\leq \gamma\|m^+\| - \left[2\gamma\|m\| - \frac{\gamma^2}{\eta}\right]$$

$$\leq \gamma\|m^+\| - 2\gamma\|m\| + \gamma\|m^+\|$$

$$= 2\gamma(\|m^+\| - \|m\|) \leq \frac{2(1-\beta)}{\beta}\gamma\|\delta\|$$

79 Thus in all cases $\min\left(\eta\|m^+\|^2, \gamma\|m^+\|\right) - \min\left(\eta\|m\|^2, \gamma\|m\|\right)$ can be upper bounded by
80 $\frac{2(1-\beta)}{\beta}\gamma\|\delta\|$. □

81 **Lemma B.3** *Suppose* $\max(\|\nabla F(x)\|, \|m^+\|, \|m\|) \geq \gamma/\eta$. *Then*

$$G(x^+, m^+) - G(x, m) \leq -\frac{2}{5}\gamma\|\nabla F(x)\| - \frac{3}{5}\frac{\gamma^2}{\eta} + \frac{12}{5\beta}\gamma\|\delta\| + \frac{AL_0 + BL_1\|\nabla F(x)\|}{2}\gamma^2 \quad (26)$$

82 **Proof:** We first write $G(x^+, m^+) - G(x, m)$ as

$$G(x^+, m^+) - G(x, m)$$
$$= \left(F(x^+) - F(x)\right) + \frac{\nu\beta}{2(1-\beta)}\left[\min\left(\eta\|m^+\|^2, \gamma\|m^+\|\right) - \min\left(\eta\|m\|^2, \gamma\|m\|\right)\right] \quad (27)$$

83 Based on Lemma B.2, we only need to bound $F(x^+) - F(x)$. We will use the $(L_0, L_1)$-smoothness
84 assumption.

$$F(x^+) - F(x)$$
$$\leq \left\langle x^+ - x, \nabla F(x)\right\rangle + \frac{AL_0 + BL_1\|\nabla F(x)\|}{2}\|x^+ - x\|^2$$
$$= -\left[\nu\min\left(\eta, \frac{\gamma}{\|m^+\|}\right)\left\langle m^+, \nabla F(x)\right\rangle + (1-\nu)\min\left(\eta, \frac{\gamma}{\|\nabla F(x)\|}\right)\left\langle\nabla F(x), \nabla F(x)\right\rangle\right]$$
$$+ \frac{AL_0 + BL_1\|\nabla F(x)\|}{2}\gamma^2$$
$$\leq \nu\left[-\frac{2}{5}\gamma\|\nabla F(x)\| - \frac{3}{5}\frac{\gamma^2}{\eta} + \left(\frac{12}{5\beta} - 1\right)\gamma\|\delta\|\right] + (1-\nu)\left(-\frac{2}{5}\gamma\|\nabla F(x)\| - \frac{3}{5}\frac{\gamma^2}{\eta} + \frac{8}{5\beta}\gamma\|\delta\|\right)$$
$$+ \frac{AL_0 + BL_1\|\nabla F(x)\|}{2}\gamma^2$$

$$(28)$$

85 Where the first inequality uses the descent inequality (Lemma A.3), the second equation follows
86 from the update rule, and the last inequality is obtained by the following two inequalities:

$$-\min\left(\eta, \frac{\gamma}{\|m^+\|}\right)\left\langle m^+, \nabla F(x)\right\rangle \leq -\frac{2}{5}\gamma\|\nabla F(x)\| - \frac{3}{5}\frac{\gamma^2}{\eta} + \left(\frac{12}{5\beta} - 1\right)\gamma\|\delta\| \quad (29)$$

$$-\min\left(\eta, \frac{\gamma}{\|\nabla F(x)\|}\right)\|\nabla F(x)\|^2 \leq -\frac{2}{5}\gamma\|\nabla F(x)\| - \frac{3}{5}\frac{\gamma^2}{\eta} + \frac{8}{5\beta}\gamma\|\delta\| \quad (30)$$

87 First we prove that (29) holds by considering the following three cases:

88 • $\|m^+\| \geq \gamma/\eta$. In this case the algorithm performs a normalized update. Then (29) follows
89 by directly using Lemma B.1 with $\mu = 2/5$:

$$-\min\left(\eta, \frac{\gamma}{\|m^+\|}\right)\left\langle m^+, \nabla F(x)\right\rangle = -\left\langle\nabla F(x), \frac{\gamma m^+}{\|m^+\|}\right\rangle$$

$$\leq -\frac{2}{5}\gamma\|\nabla F(x)\| - \frac{3}{5}\gamma\|m^+\| + \frac{7}{5}\gamma\|\delta\|$$

- $\|m^+\| < \gamma/\eta$ and $\|\nabla F(x)\| \geq \gamma/\eta$. In this case the algorithm performs an unnormalized update. We now prove $-\eta \langle \nabla F(x), m^+ \rangle \leq -\frac{2}{5}\gamma\|\nabla F(x)\| - \frac{3\gamma^2}{5\eta} + \frac{7}{5}\gamma\|\nabla F(x) - m^+\|$.

$$\eta \langle \nabla F(x), m^+ \rangle - \frac{2}{5}\gamma\|\nabla F(x)\| - \frac{3\gamma^2}{5\eta} + \frac{7}{5}\gamma\|\nabla F(x) - m^+\|$$

$$\geq \eta \langle \nabla F(x), m^+ \rangle - \frac{2}{5}\gamma\|\nabla F(x)\| - \frac{3\gamma^2}{5\eta} + \frac{7}{5}\gamma \left( \|\nabla F(x)\| - \frac{\langle \nabla F(x), m^+ \rangle}{\|\nabla F(x)\|} \right)$$

$$= \|\nabla F(x)\| \left( \gamma + \eta \frac{\langle \nabla F(x), m^+ \rangle}{\|\nabla F(x)\|} \right) - \frac{7}{5}\gamma \frac{\langle \nabla F(x), m^+ \rangle}{\|\nabla F(x)\|} - \frac{3\gamma^2}{5\eta}$$

$$\geq \frac{\gamma^2}{\eta} + \gamma \frac{\langle \nabla F(x), m^+ \rangle}{\|\nabla F(x)\|} - \frac{7}{5}\gamma \frac{\langle \nabla F(x), m^+ \rangle}{\|\nabla F(x)\|} - \frac{3\gamma^2}{5\eta}$$

$$\geq \frac{2\gamma^2}{5\eta} - \frac{2}{5}\gamma\|m^+\| \geq 0$$

- $\|m^+\| < \gamma/\eta$ and $\|\nabla F(x)\| < \gamma/\eta$. This is the most complicated case. Due to the condition in Lemma B.3, $\|m\| \geq \gamma/\eta$. In this case, the algorithm also performs an unnormalized update. We first bound $\eta \langle \nabla F(x), m \rangle$ using the same calculation as in the second case:

$$-\eta \langle \nabla F(x), m \rangle \leq -\frac{2}{5}\gamma\|m\| - \frac{3\gamma^2}{5\eta} + \frac{7}{5}\gamma\|\nabla F(x) - m\|$$

$$\leq -\frac{2}{5}\gamma\|\nabla F(x)\| - \frac{3\gamma^2}{5\eta} + \frac{7}{5\beta}\gamma\|\delta\|$$

where we use the fact that $\|\nabla F(x) - m\| = \|\delta\|/\beta$. We then bound $\eta\|\nabla F(x)\|^2$ as follows:

$$-\eta\|\nabla F(x)\|^2 \leq -2\gamma\|\nabla F(x)\| + \frac{\gamma^2}{\eta}$$

$$= -\frac{2}{5}\gamma\|\nabla F(x)\| - \frac{3\gamma^2}{5\eta} + \frac{8}{5} \left( \frac{\gamma^2}{\eta} - \gamma\|\nabla F(x)\| \right)$$

$$\leq -\frac{2}{5}\gamma\|\nabla F(x)\| - \frac{3\gamma^2}{5\eta} + \frac{8}{5}\gamma \left( \|m\| - \|\nabla F(x)\| \right)$$

$$\leq -\frac{2}{5}\gamma\|\nabla F(x)\| - \frac{3\gamma^2}{5\eta} + \frac{8}{5\beta}\|\delta\|$$

Combining the two inequalities, we obtain

$$-\langle \nabla F(x), \eta m^+ \rangle = -\eta \langle \nabla F(x), \beta m + (1 - \beta)\nabla F(x) \rangle$$

$$\leq -\frac{2}{5}\gamma\|\nabla F(x)\| - \frac{3\gamma^2}{5\eta} + \left( \frac{7}{5\beta}\beta + \frac{8}{5\beta}(1 - \beta) \right) \|\delta\|$$

$$\leq -\frac{2}{5}\gamma\|\nabla F(x)\| - \frac{3\gamma^2}{5\eta} + \left( \frac{12}{5\beta} - 1 \right) \|\delta\|$$

Thus in all cases (29) holds. We now turn to (30) which is proven in a similar fashion. Specifically, consider the following three cases:

- $\|\nabla F(x)\| \geq \gamma/\eta$. In this case

$$- \min \left( \eta, \frac{\gamma}{\|\nabla F(x)\|} \right) \|\nabla F(x)\|^2 = -\gamma\|\nabla F(x)\|^2 \leq -\frac{2}{5}\gamma\|\nabla F(x)\| - \frac{3\gamma^2}{5\eta}$$

99 • $\|\nabla F(x)\| < \gamma/\eta$ and $\|m^+\| \geq \gamma/\eta$. In this case bound $\eta\|\nabla F(x)\|^2$ the same as in the
100 third case of (29):

$$
-\min\left(\eta, \frac{\gamma}{\|\nabla F(x)\|}\right)\|\nabla F(x)\|^2 = -\eta\|\nabla F(x)\|^2
$$

$$
\leq -\frac{2}{5}\gamma\|\nabla F(x)\| - \frac{3\gamma^2}{5\eta} + \frac{8}{5}\left(\frac{\gamma^2}{\eta} - \gamma\|\nabla F(x)\|\right)
$$

$$
\leq -\frac{2}{5}\gamma\|\nabla F(x)\| - \frac{3\gamma^2}{5\eta} + \frac{8}{5}\gamma\left(\|m^+\| - \|\nabla F(x)\|\right)
$$

$$
\leq -\frac{2}{5}\gamma\|\nabla F(x)\| - \frac{3\gamma^2}{5\eta} + \frac{8}{5}\gamma\|\delta\|
$$

101 • $\|\nabla F(x)\| < \gamma/\eta$ and $\|m^+\| < \gamma/\eta$. In this case $\|m\| \geq \gamma/\eta$. Using the same calculation
102 above,

$$
-\min\left(\eta, \frac{\gamma}{\|\nabla F(x)\|}\right)\|\nabla F(x)\|^2 \leq -\frac{2}{5}\gamma\|\nabla F(x)\| - \frac{3\gamma^2}{5\eta} + \frac{8}{5\beta}\gamma\|\delta\|
$$

103 Thus (30) holds. Merging all the cases above, we finally obtain

$$
G(x^+, m^+) - G(x, m) \leq \left[-\frac{2}{5}\gamma\|\nabla F(x)\| - \frac{3}{5}\frac{\gamma^2}{\eta} + \frac{12}{5\beta}\gamma\|\delta\|\right] + \frac{AL_0 + BL_1\|\nabla F(x)\|}{2}\gamma^2
$$

$$
\tag{31}
$$

104 □

105 Now we consider all the steps $t$ which satisfy the condition in Lemma B.3, denoted as $\mathcal{S} = \{t \in$
106 $[0, T-1] : \max(\|F(x_t)\|, \|m_{t+1}\|, \|m_t\|) \geq \gamma/\eta\}$. Similarly, use $\overline{\mathcal{S}} = [0, T-1]\backslash\mathcal{S}$. Let $T_{\mathcal{S}} = |\mathcal{S}|$,
107 then $T - T_{\mathcal{S}} = |\overline{\mathcal{S}}|$.

108 **Corollary B.4** *Let set $\mathcal{S}$ and $T_{\mathcal{S}}$ be defined above. Then*

$$
\sum_{t\in\mathcal{S}} G(x_{t+1}, m_{t+1}) - G(x_t, m_t)
$$

$$
\leq \frac{12\gamma}{5\beta(1-\beta)}\|\delta_0\| + \left(\frac{12}{5(1-\beta)}AL_0 + \frac{12\gamma}{5\eta(1-\beta)}BL_1 + \frac{1}{2}AL_0\right)\gamma^2 T_{\mathcal{S}} +
$$

$$
\gamma\sum_{t\in\mathcal{S}}\left[-\frac{1}{5}(2\|\nabla F(x_t)\| + 3\frac{\gamma}{\eta}) + \frac{\gamma}{2}BL_1\|\nabla F(x_t)\| + \frac{12\gamma}{5(1-\beta)}BL_1\|\nabla F(x_t)\|\right]
$$

$$
\tag{32}
$$

109 **Proof:** Using Lemma B.3,

$$
\sum_{t\in\mathcal{S}} G(x_{t+1}, m_{t+1}) - G(x_t, m_t)
$$

$$
\leq -\sum_{t\in\mathcal{S}}\left[\frac{\gamma}{5}(2\|\nabla F(x_t)\| + 3\frac{\gamma}{\eta}) - \frac{\gamma^2}{2}BL_1\|\nabla F(x_t)\| - \frac{12}{5\beta}\gamma\|\delta_t\|\right] + \frac{\gamma^2}{2}AL_0 T_{\mathcal{S}}
$$

$$
\tag{33}
$$

110 We now focus on the summation of the term $\|\delta_t\|$. Define $S(a, b) = \nabla F(a) - \nabla F(b)$. When
111 $\|a - b\| \leq \gamma$, $\|S(a, b)\| \leq \gamma(AL_0 + BL_1\|\nabla F(b)\|)$ (see Lemma A.4). Thus we can expand

112   $\delta_t = m_{t+1} - \nabla F(x_t)$ using the recursive relation $\delta_t = \beta\delta_{t-1} + \beta S(x_{t-1}, x_t)$ as follows

$$\sum_{t\in\mathcal{S}}\|\delta_t\| = \sum_{t\in\mathcal{S}}\left\|\beta^t\delta_0 + \beta\sum_{\tau=0}^{t-1}\beta^\tau S\left(x_{t-\tau-1}, x_{t-\tau}\right)\right\|$$

$$\leq \sum_{t\in\mathcal{S}}\beta^t\|\delta_0\| + \beta\sum_{t\in\mathcal{S}}\sum_{\tau=0}^{t-1}\beta^\tau\gamma(AL_0 + BL_1\|\nabla F(x_{t-\tau})\|)$$

$$\leq \frac{1}{1-\beta}\|\delta_0\| + \frac{\beta}{1-\beta}(AL_0\gamma T_{\mathcal{S}})+$$

$$BL_1\gamma\sum_{t\in\mathcal{S}}\left(\sum_{\tau\in[1,t]\backslash\mathcal{S}}\beta^{t-\tau+1}\|\nabla F(x_\tau)\| + \sum_{\tau\in[1,t]\cap\mathcal{S}}\beta^{t-\tau+1}\|\nabla F(x_\tau)\|\right)$$

$$\leq \frac{\beta}{1-\beta}\left(\frac{\|\delta_0\|}{\beta} + AL_0\gamma T_{\mathcal{S}} + BL_1\frac{\gamma^2}{\eta}T_{\mathcal{S}} + BL_1\gamma\sum_{t\in\mathcal{S}}\|\nabla F(x_t)\|\right)$$

113   where the last inequality uses the fact that $\|\nabla F(x_\tau)\| \leq \gamma/\eta$ for all $\tau \in [1,t]\backslash\mathcal{S}$.

114   After substituting the above results into (33) we obtain

$$\sum_{t\in\mathcal{S}}G(x_{t+1}, m_{t+1}) - G(x_t, m_t)$$

$$\leq \frac{12\gamma}{5\beta(1-\beta)}\|\delta_0\| + \left(\frac{12}{5(1-\beta)}AL_0 + \frac{12\gamma}{5\eta(1-\beta)}BL_1 + \frac{1}{2}AL_0\right)\gamma^2 T_{\mathcal{S}}+ \tag{34}$$

$$\gamma\sum_{t\in\mathcal{S}}\left[-\frac{1}{5}(2\|\nabla F(x_t)\| + 3\frac{\gamma}{\eta}) + \frac{\gamma}{2}BL_1\|\nabla F(x_t)\| + \frac{12\gamma}{5(1-\beta)}BL_1\|\nabla F(x_t)\|\right]$$

115                                                                   □

116   Now we turn to the case in which $\max(\|\nabla F(x)\|, \|m^+\|, \|m\|) \leq \gamma/\eta$.

117   **Lemma B.5** *Suppose* $\max(\|\nabla F(x)\|, \|m^+\|, \|m\|) \leq \gamma/\eta$. *Then*

$$G(x^+, m^+) - G(x, m) \leq -\frac{\eta}{2}\left(c_1\|\nabla F(x)\|^2 + 2c_2\langle\nabla F(x), m\rangle + c_3\|m\|^2\right) \tag{35}$$

118   *where* $c_1 = \nu(1-\beta)(2-\beta) - L\eta(1-\beta\nu)^2 + 2(1-\nu), c_2 = \nu\beta(1-\beta) - L\eta\beta\nu(1-\beta\nu), c_3 =$
119   $\nu\beta(1+\beta) - L\eta(\beta\nu)^2$, *and* $L = AL_0 + BL_1\gamma/\eta$.

120   **Proof:**   In the case of $\|m\| \leq \gamma/\eta$, we have $\eta\|m\|^2 \leq \gamma\|m\|$, thus

$$G(x^+, m^+) - G(x, m) = (F(x^+) - F(x)) + \frac{\nu\beta\eta}{2(1-\beta)}(\|m^+\|^2 - \|m\|^2) \tag{36}$$

121   We then bound $F(x^+) - F(x)$ and $\|m^+\|^2 - \|m\|^2$. Note that $\|m^+\| \leq \gamma/\eta$ implies that the
122   algorithm performs an update without normalization. Define $L := AL_0 + BL_1\gamma/\eta$, then again by
123   descent inequality,

$$F(x^+) - F(x) \leq \langle x^+ - x, \nabla F(x)\rangle + \frac{AL_0 + BL_1\|\nabla F(x)\|}{2}\|x^+ - x\|^2$$

$$= -\left[\nu\eta\langle m^+, \nabla F(x)\rangle + (1-\nu)\eta\|\nabla F(x)\|^2\right]+$$

$$\frac{AL_0 + BL_1\|\nabla F(x)\|}{2}\eta^2\|(1-\beta\nu)\nabla(x) + \beta\nu m\|^2$$

$$\leq -\left[\nu\eta\langle m^+, \nabla F(x)\rangle + (1-\nu)\eta\|\nabla F(x)\|^2\right] + \frac{L}{2}\eta^2\|(1-\beta\nu)\nabla F(x) + \beta\nu m\|^2$$

$$\overset{Rearranging}{\leq} -\left[\nu(1-\beta)\eta + (1-\nu)\eta - \frac{L}{2}\eta^2(1-\beta\nu)^2\right]\|\nabla F(x)\|^2$$

$$- \left[\nu\beta\eta - L\eta^2(\beta\nu)(1-\beta\nu)\right]\langle\nabla F(x), m\rangle + \frac{L}{2}\eta^2\beta^2\nu^2\|m\|^2$$

$$\tag{37}$$

124 Since

$$\left\|m^+\right\|^2 - \|m\|^2 = (1-\beta)^2\|\nabla F(x)\|^2 - (1+\beta)(1-\beta)\|m\|^2 + 2\beta(1-\beta)\langle\nabla F(x), m\rangle \quad (38)$$

by definition of the energy function, we have

$$G(x^+, m^+) - G(x, m) \le -\frac{\eta}{2}\left(c_1\|\nabla F(x)\|^2 + 2c_2\langle\nabla F(x), m\rangle + c_3\|m\|^2\right)$$

125 where $c_1 = \nu(1-\beta)(2-\beta) - L\eta(1-\beta\nu)^2 + 2(1-\nu), c_2 = \nu\beta(1-\beta) - L\eta\beta\nu(1-\beta\nu), c_3 =$
126 $\nu\beta(1+\beta) - L\eta(\beta\nu)^2$. $\qquad\square$

**Lemma B.6** *Let $c_1, c_2, c_3$ and L be defined in Lemma B.5. If $L\eta \le 1$, then the matrix*

$$H = \begin{pmatrix} [c_1 - (1-\nu\beta)]I_d & c_2 I_d \\ c_2 I_d & (c_3 - \nu\beta)I_d \end{pmatrix}$$

127 *is symmetric and positive semi-definite, where $I_d$ is the $d \times d$ identity matrix.*

**Proof:** In fact we only need to consider the case when $d = 1$, because the eigenvalues of $H_{2d\times 2d}$ can only be those that appears in $H_{2\times 2}$ $(d = 1)$. Denote two eigenvalues be $\lambda_1, \lambda_2$ when $d = 1$. A direct calculation shows that

$$\lambda_1\lambda_2 = \det H = [c_1 - (1-\nu\beta)](c_3 - \nu\beta) - c_2^2 = \nu(1-\nu)\beta^2(1 - L\eta)$$

$$\lambda_1 + \lambda_2 = c_1 + c_3 - 1 = (1-\nu\beta)^2(1-L\eta) + (\nu\beta)^2(1-L\eta) + 2\beta^2\nu(1-\nu)$$

128 If $L\eta \le 1$, then $\lambda_1\lambda_2 \ge 0$ and $\lambda_1 + \lambda_2 \ge 0$, which is equivalent to the semi-definiteness of $H$. $\quad\square$

129 **Corollary B.7** *Suppose $\max(\|\nabla F(x)\|, \|m\|, \|m^+\|) \le \gamma/\eta$. If $L\eta \le 1$, Then*

$$G(x^+, m^+) - G(x, m) \le -\frac{\eta}{2}(1-\nu\beta)\|\nabla F(x)\|^2 - \frac{\eta}{2}\nu\beta\|m\|^2 \quad (39)$$

130 **Proof:** Let $H$ be defined in Lemma B.6. The result of Lemma B.5 can be written in a matrix form:
131

$$G(x^+, m^+) - G(x, m) \le -\frac{\eta}{2}(1-\nu\beta)\|\nabla F(x)\|^2 - \frac{\eta}{2}\nu\beta\|m\|^2 - \frac{\eta}{2}\left(\nabla F(x)^T, m^T\right) H \left(\nabla F(x)^T, m^T\right)^T$$
$$(40)$$

132 Using the fact that $H$ is positive semi-definite, we obtain the desired result. $\qquad\square$

133 Note that the amount of descent in Corollary B.7 is small in terms of $\|\nabla F(x)\|$ if $\beta$ and $\nu$ are close
134 to 1. We now try to convert the term $\|m\|$ into $\|\nabla F(x)\|$, which is stated in the following lemma.

135 **Lemma B.8** *Suppose $AL_0\eta \le c_1(1-\beta)$ and $BL_1\gamma \le c_3(1-\beta)$ for some constant $c_1$ and $c_3$. Let*
136 $m_0 = \nabla F(x_0)$ *for simplicity. Let set $\mathcal{S}$ and $\overline{\mathcal{S}}$ be defined above. Then*

$$\sum_{t\in\overline{\mathcal{S}}}\|m_t\| \ge \frac{1}{1+c_1}\sum_{t\in\overline{\mathcal{S}}}((1-c_1(1-\nu\beta)-c_3)\|\nabla F(x_t)\|)$$
$$-\frac{1}{1-\beta}\sum_{t\in\mathcal{S}}(AL_0 + BL_1\|\nabla F(x_t)\|)\gamma \quad (41)$$

137 **Proof:** For any $t \ge 1$, we have

$$\|m_t - \nabla F(x_t)\| \le \|m_t - \nabla F(x_{t-1})\| + \|\nabla F(x_{t-1}) - \nabla F(x_t)\|$$
$$\le \beta\|m_{t-1} - \nabla F(x_{t-1})\| + (AL_0 + BL_1\|\nabla F(x_{t-1})\|)\times$$
$$\left(\nu\min\left(\eta, \frac{\gamma}{\|m_t\|}\right)\|m_t\| + (1-\nu)\min\left(\eta, \frac{\gamma}{\|\nabla F(x_{t-1})\|}\right)\|\nabla F(x_{t-1})\|\right)$$
$$(42)$$

138 where the last inequality follows by Corollary A.4. Applying (42) recursively, we obtain

$$\|m_t - \nabla F(x_t)\| \le \sum_{\tau=1}^{t}\beta^{t-\tau}(AL_0 + BL_1\|\nabla F(x_{\tau-1})\|)\times$$
$$\left(\nu\min\left(\eta, \frac{\gamma}{\|m_\tau\|}\right)\|m_\tau\| + (1-\nu)\min\left(\eta, \frac{\gamma}{\|\nabla F(x_{\tau-1})\|}\right)\|\nabla F(x_{\tau-1})\|\right)$$
$$(43)$$

139  Therefore,

$$\sum_{t=0}^{T-1} \|m_t - \nabla F(x_t)\|$$

$$\leq \frac{1}{1-\beta} \sum_{t=0}^{T-1} (AL_0 + BL_1 \|\nabla F(x_t)\|) \left( \nu \min\left( \eta, \frac{\gamma}{\|m_{t+1}\|} \right) \|m_{t+1}\| + (1-\nu) \min\left( \eta, \frac{\gamma}{\|\nabla F(x_t)\|} \right) \|\nabla F(x_t)\| \right)$$

$$\leq \frac{1}{1-\beta} \left( \sum_{t=0}^{T-1} BL_1 \gamma \|\nabla F(x_t)\| + \sum_{t\in\mathcal{S}} AL_0 \gamma + \sum_{t\in\overline{\mathcal{S}}} AL_0 \eta \left( (1-\nu)\|\nabla F(x_t)\| + \nu\|m_{t+1}\| \right) \right)$$

(44)

140  Therefore we obtain

$$\sum_{t=0}^{T-1} \|m_t - \nabla F(x_t)\|$$

$$\leq \frac{1}{1-\beta} \left( \sum_{t\in\mathcal{S}} (AL_0 + BL_1\|\nabla F(x_t)\|)\gamma + \sum_{t\in\overline{\mathcal{S}}} \left[ AL_0\nu\eta\|m_{t+1}\| + (BL_1\gamma + AL_0(1-\nu)\eta)\|\nabla F(x_t)\| \right] \right)$$

$$\leq \frac{1}{1-\beta} \left( \sum_{t\in\mathcal{S}} (AL_0 + BL_1\|\nabla F(x_t)\|)\gamma \right) +$$

$$\frac{1}{1-\beta} \left( \sum_{t\in\overline{\mathcal{S}}} AL_0\nu\eta\beta\|m_t\| + (AL_0\eta(1-\nu\beta) + BL_1\gamma)\|\nabla F(x_t)\| \right)$$

$$\leq \frac{1}{1-\beta} \left( \sum_{t\in\mathcal{S}} (AL_0 + BL_1\|\nabla F(x_t)\|)\gamma \right) + \left( \sum_{t\in\overline{\mathcal{S}}} (c_1(1-\nu\beta) + c_3)\|\nabla F(x_t)\| + c_1\nu\beta\|m_t\| \right)$$

(45)

141  Using $\|m_t\| \geq \|\nabla F(x_t)\| - \|m_t - \nabla F(x_t)\|$ and some straightforward calculation, we obtain

$$(1+c_1)\sum_{t\in\overline{\mathcal{S}}} \|m_t\| \geq \left( \sum_{t\in\overline{\mathcal{S}}} (1 - c_1(1-\nu\beta) - c_3)\|\nabla F(x_t)\| \right)$$

$$- \frac{1}{1-\beta} \left( \sum_{t\in\mathcal{S}} (AL_0 + BL_1\|\nabla F(x_t)\|)\gamma \right)$$

(46)

142                                                                                                $\square$

143  Now we are ready to prove the main theorem.

**Theorem B.9** *Let $F^*$ be the optimal value, and $\Delta = F(x_0) - F^*$. Assume $m_0 = \nabla F(x_0)$ for simplicity. If $\gamma \leq \frac{1-\beta}{10BL_1}$ and $\eta \leq \frac{1-\beta}{10AL_0}$, where constants $A = 1 + e^{1/10} - 10(e^{1/10} - 1) < 1.06$, $B = 10(e^{1/10} - 1) < 1.06$, and $\varepsilon < \frac{\gamma}{5\eta}$, then*

$$\frac{1}{T}\sum_{t=1}^{T} \|\nabla F(x_t)\| \leq 2\varepsilon$$

144  *as long as*

$$T \geq \frac{3}{\varepsilon^2 \eta}\Delta$$

(47)

145  **Proof:**  By calculating $L\eta = AL_0\eta + BL_1\gamma \leq (1-\beta)/5 < 1$, we can use Corollary B.7. Taking
146  summation of the inequality (39) over steps $t \in \overline{\mathcal{S}} = [0, T-1]\backslash\mathcal{S}$, we obtain

$$\sum_{t\in\overline{\mathcal{S}}} G(x_{t+1}, m_{t+1}) - G(x_t, m_t) \leq -\frac{\eta}{2}\sum_{t\in\overline{\mathcal{S}}} \left( (1-\nu\beta)\|\nabla F(x_t)\|^2 + \nu\beta\|m_t\|^2 \right)$$

(48)

147 Combining (48) and (32) in Corollary B.4 we obtain

$$
\begin{aligned}
G(x_T, m_T) - G(x_0, m_0) &= \sum_{t=0}^{T-1} G(x_{t+1}, m_{t+1}) - G(x_t, m_t) \\
&\leq -\frac{\eta}{2} \sum_{t \in \overline{\mathcal{S}}} \left( (1 - \nu\beta) \|\nabla F(x_t)\|^2 + \nu\beta \|m_t\|^2 \right) + \\
&\quad \frac{12\gamma}{5\beta(1-\beta)} \|\delta_0\| + \left( \frac{12}{5(1-\beta)} AL_0 + \frac{12\gamma}{5\eta(1-\beta)} BL_1 + \frac{1}{2} AL_0 \right) \gamma^2 T_{\mathcal{S}} + \\
&\quad \gamma \sum_{t \in \mathcal{S}} \left[ -\frac{1}{5} (2\|\nabla F(x_t)\| + 3\frac{\gamma}{\eta}) + \left( \frac{1}{2} + \frac{12}{5(1-\beta)} \right) BL_1 \gamma \|\nabla F(x_t)\| \right]
\end{aligned}
\tag{49}
$$

148 By the assumption

$$
\gamma \leq \frac{1-\beta}{10BL_1}, \quad \eta \leq \frac{1-\beta}{10AL_0}
\tag{50}
$$

149 we have $AL_0\eta \leq (1-\beta)/10$ and $BL_1\gamma \leq (1-\beta)/10$. Using Lemma B.14 we have

$$
\sum_{t \in \overline{\mathcal{S}}} \|m_t\| \geq \frac{8}{11} \sum_{t \in \overline{\mathcal{S}}} \|\nabla F(x_t)\| - \frac{1}{1-\beta} \left( \sum_{t \in \mathcal{S}} (AL_0 + BL_1 \|\nabla F(x_t)\|) \gamma \right)
\tag{51}
$$

150 Therefore by standard inequality $x^2 \geq 2\varepsilon x - \varepsilon^2$ and (49) we obtain

$$
\begin{aligned}
G(x_0, m_0) &- G(x_T, m_T) \\
&\geq \frac{\eta}{2} \sum_{t \in \overline{\mathcal{S}}} \left( (1-\nu\beta)\|\nabla F(x_t)\|^2 + 2\nu\beta\varepsilon\|m_t\| - \nu\beta\varepsilon^2 \right) \\
&\quad + \left( \frac{3}{5}\frac{\gamma^2}{\eta} - \left( \frac{12}{5(1-\beta)} AL_0 + \frac{12\gamma}{5\eta(1-\beta)} BL_1 + \frac{1}{2} AL_0 \right) \gamma^2 \right) T_{\mathcal{S}} \\
&\quad + \gamma \left( \frac{2}{5} - \left( \frac{1}{2} + \frac{12}{5(1-\beta)} \right) BL_1 \gamma \right) \sum_{t \in \mathcal{S}} \|\nabla F(x_t)\| \\
&\geq \sum_{t \in \mathcal{S}} U(x_t) + \sum_{t \in \overline{\mathcal{S}}} V(x_t)
\end{aligned}
\tag{52}
$$

151 Where

$$
\begin{aligned}
U(x) &:= \left( \frac{3}{5}\frac{\gamma^2}{\eta} - \left( \frac{12}{5(1-\beta)} AL_0 + \frac{12\gamma}{5\eta(1-\beta)} BL_1 + \frac{1}{2} AL_0 \right) \gamma^2 - \frac{\nu\beta}{1-\beta} AL_0 \varepsilon\gamma\eta \right) \\
&\quad + \gamma \left( \frac{2}{5} - \left( \frac{1}{2} + \frac{12}{5(1-\beta)} \right) BL_1 \gamma - \frac{\nu\beta}{1-\beta} \varepsilon\eta BL_1 \right) \|\nabla F(x)\| \\
V(x) &:= \frac{\eta}{2}(1 - \nu\beta)\|\nabla F(x)\|^2 + \frac{8}{11} \nu\beta\varepsilon\eta\|\nabla F(x)\| - \frac{1}{2}\nu\beta\varepsilon^2\eta
\end{aligned}
\tag{53}
$$

152 We now simplify $U(x)$. Let $\varepsilon \leq \frac{\gamma}{5\eta}$. By (50) we have

$$
\begin{aligned}
\frac{2}{5} - \left( \frac{1}{2} + \frac{12}{5(1-\beta)} \right) BL_1 \gamma - \frac{\nu\beta}{1-\beta} \varepsilon\eta BL_1 &\geq \frac{2}{5} - \frac{12}{50} - \frac{1}{20} \geq \frac{1}{10} \\
\frac{3}{5} - \frac{12\gamma}{5(1-\beta)} BL_1 &\geq \frac{3}{10}
\end{aligned}
\tag{54}
$$

153 Therefore

$$
\begin{aligned}
U(x) &\geq \frac{3}{10}\frac{\gamma^2}{\eta} - \left( \frac{12}{5(1-\beta)} + \frac{1}{2} \right) AL_0 \gamma^2 - \frac{\nu\beta}{1-\beta} AL_0 \varepsilon\gamma\eta + \frac{1}{10}\gamma\|\nabla F(x)\| \\
&\geq \left( \frac{3}{5(1-\beta)} - \frac{1}{2} \right) AL_0 \gamma^2 - \frac{\nu\beta}{1-\beta} AL_0 \varepsilon\gamma\eta + \frac{1}{10}\gamma\|\nabla F(x)\| \\
&\geq \frac{1}{10(1-\beta)} AL_0 \gamma^2 + \frac{1}{10}\gamma\|\nabla F(x)\|
\end{aligned}
\tag{55}
$$

We can also bound $V(x)$ as follows:

$$
\begin{aligned}
V(x) &\geq (1 - \nu\beta)\varepsilon\eta\|\nabla F(x)\| - \frac{\eta}{2}(1 - \nu\beta)\varepsilon^2 + \frac{8}{11}\nu\beta\varepsilon\eta\|\nabla F(x)\| - \frac{1}{2}\nu\beta\varepsilon^2\eta \\
&\geq \frac{1}{2}\varepsilon\eta\|\nabla F(x)\| - \frac{1}{2}\varepsilon^2\eta
\end{aligned}
\tag{56}
$$

Since $\varepsilon < \dfrac{\gamma}{5\eta}$, we have $U(x) \geq V(x)$. Therefore by (52) and Lemma A.5 we have

$$
\begin{aligned}
T\sum_{t=0}^{T-1}\frac{1}{2}\varepsilon\eta\left(\|\nabla F(x)\| - \varepsilon\right) &\leq \Delta + \frac{\beta}{2(1-\beta)}\min\{\eta\|\nabla F(x_0)\|^2, \gamma\|\nabla F(x_0)\|\} \\
&\leq \Delta + \frac{4\beta}{1-\beta}\Delta\max\{L_0\eta, L_1\gamma\} \\
&\leq \frac{7}{5}\Delta
\end{aligned}
\tag{57}
$$

Thus

$$
\frac{1}{T}\sum_{t=0}^{T-1}\|\nabla F(x_t)\| \leq 2\varepsilon
\tag{58}
$$

as long as

$$
T > \frac{3}{\varepsilon^2\eta}\Delta
\tag{59}
$$

$\square$

## B.2   Proof of Theorem 3.2

We now prove the stochastic case. As before, to simplify the notation we write the update formula as

$$
\begin{aligned}
m^+ &= \beta m + (1 - \beta)\nabla f(x, \xi) \\
x^+ &= x - \left(\nu\min\left(\eta, \frac{\gamma}{\|m^+\|}\right)m^+ + (1 - \nu)\min\left(\eta, \frac{\gamma}{\|\nabla f(x, \xi)\|}\right)\nabla f(x, \xi)\right)
\end{aligned}
\tag{60}
$$

when analyzing a single iteration. The error between $m^+$ and $\nabla F(x)$ is denoted as $\delta = m^+ - \nabla F(x)$. We define the true momentum $\tilde{m}$ as follows:

$$
\tilde{m}^+ = \beta\tilde{m} + (1 - \beta)\nabla F(x)
\tag{61}
$$

where $\tilde{m}_0 = m_0$. Similarly, the error between $\tilde{m}^+$ and $\nabla F(x)$ is denoted as $\tilde{\delta} = \tilde{m}^+ - \nabla F(x)$.

In stochastic case, we define the energy function to be

$$
G(x, \tilde{m}) = F(x) + \frac{\nu\beta}{2(1-\beta)}\min\left(\eta\|\tilde{m}\|^2, \gamma\|\tilde{m}\|\right)
\tag{62}
$$

The only change is that we use the true momentum $\tilde{m}$ instead of stochastic momentum $m$. Note that Lemma B.1 and Lemma B.2 can still be used in stochastic case. The momentum $m$ and error $\delta$ in Lemma B.2 will be changed to $\tilde{m}$ and $\tilde{\delta}$ respectively.

Suppose $\gamma \leq c/L_1$ for some constant $c$, and we denote $A = 1 + e^c - \frac{e^c - 1}{c}$ and $B = \frac{e^c - 1}{c}$, just the same as in the descent inequality (Lemma A.3). When $\gamma \leq \frac{1-\beta}{50L_1}\varepsilon \leq \frac{1}{500L_1}$ (in Theorem 3.2), we can take $c = 1/500$ and $A = B = 1.002$.

**Lemma B.10** *The difference between $m$ and $\tilde{m}$ satisfies:*

$$
\|m^+ - \tilde{m}^+\| \leq \sigma
\tag{63}
$$

*Furthermore, in expectation*

$$
\mathbb{E}\|m^+ - \tilde{m}^+\|^2 \leq \frac{1-\beta}{1+\beta}\sigma
\tag{64}
$$

174 **Proof:** By expanding $m_{t+1}$ and $\tilde{m}_{t+1}$, we get

$$\|m_{t+1} - \tilde{m}_{t+1}\| = (1-\beta) \left\| \sum_{\tau=0}^{t} \beta^{t-\tau}(\nabla f(x_\tau, \xi_\tau) - \nabla F(x_\tau)) \right\|$$

$$\le (1-\beta) \sum_{\tau=0}^{t} \beta^{t-\tau} \|\nabla f(x_\tau, \xi_\tau) - \nabla F(x_\tau)\| \quad (65)$$

$$\le (1-\beta) \sum_{\tau=0}^{t} \beta^{t-\tau} \sigma \le \sigma$$

Furthermore, using the noise assumption, for different time steps $t, t'$, we have

$$\mathbb{E}[\langle \nabla f(x_t, \xi_t) - \nabla F(x_t), \nabla f(x_{t'}, \xi_{t'}) - \nabla F(x_{t'})\rangle] = 0$$

175 Therefore

$$\mathbb{E}[\|m_{t+1} - \tilde{m}_{t+1}\|^2] = \mathbb{E}\left[ \sum_{\tau=0}^{t}(1-\beta)^2 \beta^{2(t-\tau)} \|\nabla f(x_\tau, \xi_\tau) - \nabla F(x_\tau)\|^2\right] \le \frac{1-\beta}{1+\beta}\sigma^2 \quad (66)$$

176 $\qquad\qquad\qquad\qquad\qquad\qquad\qquad\qquad\qquad\qquad\qquad\qquad\qquad\qquad\qquad\qquad\qquad\qquad\quad \square$

177 **Lemma B.11** *Suppose* $\max(5\|\nabla F(x)\|/4, \|m^+\|, \|\tilde{m}\|) \ge \gamma/\eta$. *Then*

$$G(x^+, \tilde{m}^+) - G(x, \tilde{m})$$

$$\le -\frac{4}{5} \times \frac{2\gamma}{5}\|\nabla F(x)\| - \frac{16}{25} \times \frac{3\gamma^2}{5\eta} + \frac{\gamma^2}{2}(AL_0 + BL_1\|\nabla F(x)\|) + \frac{12}{5\beta}\gamma\|\tilde{\delta}\|$$

$$- \nu\eta\langle \nabla F(x), m^+ - \tilde{m}^+\rangle - (1-\nu)\eta\langle \nabla F(x), \nabla f(x,\xi) - \nabla F(x)\rangle + \left(\eta\|\nabla F(x)\| + \frac{7}{5}\gamma\right)\sigma$$
$$(67)$$

178 **Proof:** Based on Lemma B.2, we only need to bound $F(x^+) - F(x)$. We use the $(L_0, L_1)$-smooth
179 condition:

$$F(x^+) - F(x) \le \langle \nabla F(x), x^+ - x\rangle + \frac{\gamma^2}{2}(AL_0 + BL_1\|\nabla F(x)\|) \quad (68)$$

180 Now we bound $\langle \nabla F(x), x^+ - x\rangle$. The calculation is similar to the deterministic setting. We first
181 bound $-\min\left(\eta, \frac{\gamma}{\|m^+\|}\right)\langle m^+, \nabla F(x)\rangle$. Consider the following three cases, all of which are analo-
182 gous to the proof of Lemma B.3:

183 $\qquad\bullet$ $\|m^+\| \ge \gamma/\eta$. The algorithm performs a normalized update. We have

$$-\frac{\gamma}{\|m^+\|}\langle m^+, \nabla F(x)\rangle \le -\frac{2}{5}\gamma\|\nabla F(x)\| - \frac{3}{5}\gamma\|m^+\| + \frac{7}{5}\gamma\|\delta\|$$

184 $\qquad\bullet$ $\|m^+\| < \gamma/\eta$ and $\|\nabla F(x)\| \ge 4\gamma/5\eta$. The algorithm performs an unnormalized update.
185 $\qquad$ We have

$$-\eta\langle \nabla F(x), m^+\rangle \le -\frac{4}{5} \times \frac{2}{5}\gamma\|\nabla F(x)\| - \frac{16}{25} \times \frac{3\gamma^2}{5\eta} + \frac{4}{5} \times \frac{7}{5}\gamma\|\delta\|$$

186 $\qquad\bullet$ $\|m^+\| < \gamma/\eta$ and $\|\nabla F(x)\| < 4\gamma/5\eta$. In this case $\|\tilde{m}\| \ge \gamma/\eta$. The algorithm performs
187 $\qquad$ an unnormalized update. We have

$$-\eta\langle \nabla F(x), \tilde{m}^+\rangle \le -\frac{2}{5}\gamma\|\nabla F(x)\| - \frac{3\gamma^2}{5\eta} + \left(\frac{12}{5\beta} - 1\right)\gamma\|\tilde{\delta}\|$$

188 Therefore in all the cases, we have

$$-\min\left(\eta, \frac{\gamma}{\|m^+\|}\right)\langle m^+, \nabla F(x)\rangle \le -\frac{4}{5} \times \frac{2}{5}\gamma\|\nabla F(x)\| - \frac{16}{25} \times \frac{3\gamma^2}{5\eta} + \left(\frac{12}{5\beta} - 1\right)\gamma\|\tilde{\delta}\|$$

$$- \eta\langle \nabla F(x), m^+ - \tilde{m}^+\rangle + \left(\eta\|\nabla F(x)\| + \frac{7}{5}\gamma\right)\sigma \quad (69)$$

189 where (69) uses the following two inequalities which can be obtained by Lemma B.10:

$$\|\delta\| \leq \|\tilde{\delta}\| + \sigma \tag{70}$$

$$-\langle \nabla F(x), m^+ - \tilde{m}^+ \rangle \leq \|\nabla F(x)\| \sigma \tag{71}$$

190 We next bound $-\min\left(\eta, \frac{\gamma}{\|\nabla f(x,\xi)\|}\right) \langle \nabla f(x,\xi), \nabla F(x) \rangle$. Consider the following cases, all of
191 which are analogous to the proof of Lemma B.3:

192 • $\|\nabla f(x,\xi)\| \geq \gamma/\eta$. In this case we can use Lemma B.1 with $\mu = 2/5$:

$$
\begin{aligned}
& -\min\left(\eta, \frac{\gamma}{\|\nabla f(x,\xi)\|}\right) \langle \nabla f(x,\xi), \nabla F(x) \rangle \\
&= -\gamma \frac{\langle \nabla f(x,\xi), \nabla F(x) \rangle}{\|\nabla f(x,\xi)\|} \\
&\leq \gamma\left(-\frac{2}{5}\|\nabla F(x)\| - \frac{3}{5}\|\nabla f(x,\xi)\| + \frac{7}{5}\|\nabla F(x) - \nabla f(x,\xi)\|\right) \\
&\leq \gamma\left(-\frac{2}{5}\|\nabla F(x)\| - \frac{3\gamma}{5\eta} + \frac{7}{5}\sigma\right)
\end{aligned}
\tag{72}
$$

193 • $\|\nabla f(x,\xi)\| < \gamma/\eta$. In this case

$$
\begin{aligned}
& -\min\left(\eta, \frac{\gamma}{\|\nabla f(x,\xi)\|}\right) \langle \nabla f(x,\xi), \nabla F(x) \rangle \\
&= -\eta \langle \nabla f(x,\xi), \nabla F(x) \rangle \\
&= -\eta\|\nabla F(x)\|^2 - \eta \langle \nabla f(x,\xi) - \nabla F(x), \nabla F(x) \rangle
\end{aligned}
\tag{73}
$$

194 We now bound $-\eta\|\nabla F(x)\|^2$. If $\|\nabla F(x)\| \geq \frac{4\gamma}{5\eta}$, then $-\eta\|\nabla F(x)\|^2 \leq -\frac{4}{5}\gamma\|\nabla F(x)\|$.
195 If $\|\nabla F(x)\| < \frac{4\gamma}{5\eta}$ and $\|m^+\| \geq \frac{4\gamma}{5\eta}$, then using the same calculation as in the deterministic
196 case,

$$
\begin{aligned}
-\eta\|\nabla F(x)\|^2 &\leq -\frac{2}{5} \times \frac{4}{5}\gamma\|\nabla F(x)\| - \frac{16}{25} \times \frac{3\gamma^2}{5\eta} + \frac{4}{5} \times \frac{8}{5}\gamma\left(\|m^+\| - \|\nabla F(x)\|\right) \\
&\leq -\frac{4}{5} \times \frac{2}{5}\gamma\|\nabla F(x)\| - \frac{16}{25} \times \frac{3\gamma^2}{5\eta} + \frac{7}{5}\gamma\left(\|\tilde{\delta}\| + \sigma\right)
\end{aligned}
$$

197 If $\|\nabla F(x)\| < \frac{4\gamma}{5\eta}$ and $\|m^+\| < \frac{4\gamma}{5\eta}$, then $\|\tilde{m}\| \geq \gamma/\eta$. Using the same calculation we have

$$
\begin{aligned}
-\eta\|\nabla F(x)\|^2 &\leq -\frac{2}{5}\gamma\|\nabla F(x)\| - \frac{3\gamma^2}{5\eta} + \frac{8}{5}\gamma\left(\|\tilde{m}\| - \|\nabla F(x)\|\right) \\
&\leq -\frac{2}{5}\gamma\|\nabla F(x)\| - \frac{3\gamma^2}{5\eta} + \frac{8}{5\beta}\gamma\|\tilde{\delta}\|
\end{aligned}
$$

198 Therefore in all the cases we have

$$
\begin{aligned}
-\min\left(\eta, \frac{\gamma}{\|\nabla f(x,\xi)\|}\right) \langle \nabla f(x,\xi), \nabla F(x) \rangle \leq & -\frac{4}{5} \times \frac{2}{5}\gamma\|\nabla F(x)\| - \frac{16}{25} \times \frac{3\gamma^2}{5\eta} + \frac{8}{5\beta}\gamma\|\tilde{\delta}\| \\
& -\eta \langle \nabla F(x), \nabla f(x,\xi) - \nabla F(x) \rangle + \left(\eta\|\nabla F(x)\| + \frac{7}{5}\gamma\right)\sigma
\end{aligned}
\tag{74}
$$

199 we finally obtain

$$
\begin{aligned}
& G(x^+, \tilde{m}^+) - G(x, \tilde{m}) \\
&\leq -\frac{4}{5} \times \frac{2\gamma}{5}\|\nabla F(x)\| - \frac{16}{25} \times \frac{3\gamma^2}{5\eta} + \frac{\gamma^2}{2}(AL_0 + BL_1\|\nabla F(x)\|) + \frac{12}{5\beta}\gamma\|\tilde{\delta}\| \\
& -\nu\eta \langle \nabla F(x), m^+ - \tilde{m}^+ \rangle - (1-\nu)\eta \langle \nabla F(x), \nabla f(x,\xi) - \nabla F(x) \rangle + \left(\eta\|\nabla F(x)\| + \frac{7}{5}\gamma\right)\sigma
\end{aligned}
\tag{75}
$$

200 $\qquad\qquad\qquad\qquad\qquad\qquad\qquad\qquad\qquad\qquad\qquad\qquad\qquad\qquad\qquad\qquad\quad$ $\square$

201 Let $\mathcal{S} = \{t \in [0, T-1] : \max(5\|F(x_t)\|/4, \|m_{t+1}\|, \|\tilde{m}_t\|) \geq \gamma/\eta\}$ and $\overline{\mathcal{S}} = [0, T-1]\backslash\mathcal{S}$. Let
202 $T_{\mathcal{S}} = |\mathcal{S}|$, then $T - T_{\mathcal{S}} = |\overline{\mathcal{S}}|$. Parallel to Corollary B.4, we directly have the following corollary.

203 **Corollary B.12** *Let set $\mathcal{S}$ and $T_{\mathcal{S}}$ be defined above. Then*

$$
\sum_{t\in\mathcal{S}} G(x_{t+1}, m_{t+1}) - G(x_t, m_t)
$$
$$
\leq \frac{12\gamma}{5\beta(1-\beta)}\|\tilde{\delta}_0\| + \left(\frac{12}{5(1-\beta)}AL_0 + \frac{12\gamma}{5\eta(1-\beta)}BL_1 + \frac{1}{2}AL_0\right)\gamma^2 T_{\mathcal{S}}
$$
$$
- \eta\sum_{t\in\mathcal{S}}(\nu\langle\nabla F(x_t), m_{t+1} - \tilde{m}_{t+1}\rangle + (1-\nu)\langle\nabla F(x_t), \nabla f(x_t, \xi_t) - \nabla F(x_t)\rangle) +
$$
$$
\gamma\sum_{t\in\mathcal{S}}\left[-\left(\left(\frac{4}{5}\times\frac{2}{5} - \frac{\eta}{\gamma}\sigma\right)\|\nabla F(x_t)\| + \left(\frac{16}{25}\times\frac{3\gamma}{5\eta} - \frac{7}{5}\sigma\right)\right) + \frac{\gamma}{2}BL_1\|\nabla F(x_t)\| + \frac{12\gamma}{5(1-\beta)}BL_1\|\nabla F(x_t)\|\right]
$$
$$(76)$$

204 Next we turn to the case in which $\max(5\|\nabla F(x)\|/4, \|m^+\|, \|\tilde{m}\|) \leq \gamma/\eta$.

205 **Lemma B.13** *Assume $\max(5\|\nabla F(x)\|/4, \|m^+\|, \|\tilde{m}\|) \leq \gamma/\eta$, and $\gamma/\eta = 5\sigma$. If $AL_0\eta \leq 1$, then*
206

$$
G(x^+, \tilde{m}^+) - G(x, \tilde{m})
$$
$$
\leq -\frac{\eta}{2}(1-\nu\beta)\|\nabla F(x)\|^2 - \frac{\eta}{2}\nu\beta\|\tilde{m}\|^2 + \frac{\gamma^2}{2}BL_1\|\nabla F(x)\|
$$
$$
- \nu\eta\langle\nabla F(x), m^+ - \tilde{m}^+\rangle - (1-\nu)\eta\langle\nabla F(x), \nabla f(x, \xi) - \nabla F(x)\rangle
$$
$$
+ \eta^2 AL_0\sigma\|\nu\tilde{m}^+ + (1-\nu)\nabla F(x)\| + \frac{1}{2}\eta^2 AL_0\|\nu(m^+ - \tilde{m}^+) + (1-\nu)(\nabla f(x, \xi) - \nabla F(x))\|^2
$$
$$(77)$$

207 *where $c_1 = \nu(1-\beta)(2-\beta) - AL_0\eta(1-\beta\nu)^2 + 2(1-\nu), c_2 = \nu\beta(1-\beta) - AL_0\eta\beta\nu(1-\beta\nu), c_3 =$*
208 *$\nu\beta(1+\beta) - AL_0\eta(\beta\nu)^2$. $c_1 = (1-\beta)[2-\beta - AL_0\eta(1-\beta)], c_2 = \beta[1-\beta - AL_0\eta(1-\beta)]$ and*
209 *$c_3 = \beta(1+\beta - AL_0\eta\beta)$.*

210 **Proof:** Because $\|\nabla f(x, \xi)\| \leq 4\gamma/5\eta + \sigma = \gamma/\eta$ and $\|m^+\| \leq \gamma/\eta$, the algorithm performs an
211 unnormalized update. The proof is similar to the one in Lemma B.5 except for bounding the term
212 $F(x^+) - F(x)$.

$$
F(x^+) - F(x)
$$
$$
\leq -\langle\nabla F(x), \nu\eta m^+ + (1-\nu)\eta\nabla f(x, \xi)\rangle + \frac{\eta^2}{2}(AL_0 + BL_1\|\nabla F(x)\|)\|\nu m^+ + (1-\nu)\nabla f(x, \xi)\|^2
$$
$$
\leq -\nu\eta\langle\nabla F(x), \tilde{m}^+\rangle - \nu\eta\langle\nabla F(x), m^+ - \tilde{m}^+\rangle
$$
$$
- (1-\nu)\eta\langle\nabla F(x), \nabla F(x)\rangle - (1-\nu)\eta\langle\nabla F(x), \nabla f(x, \xi) - \nabla F(x)\rangle
$$
$$
+ \frac{\eta^2}{2}AL_0\left(\|\nu\tilde{m}^+ + (1-\nu)\nabla F(x)\|^2 + \|\nu(m^+ - \tilde{m}^+) + (1-\nu)(\nabla f(x, \xi) - \nabla F(x))\|^2\right)
$$
$$
+ \eta^2 AL_0\sigma\|\nu\tilde{m}^+ + (1-\nu)\nabla F(x)\| + \frac{\eta^2}{2}BL_1\|\nabla F(x)\|\frac{\gamma^2}{\eta^2}
$$
$$(78)$$

213 For bounding term $-\nu\eta\langle\nabla F(x), \tilde{m}^+\rangle - (1-\nu)\eta\langle\nabla F(x), \nabla F(x)\rangle + \frac{\eta^2}{2}AL_0\|\nu\tilde{m}^+ + (1-$
214 $\nu)\nabla F(x)\|^2$ that is not related to noise, the subsequent steps are the same as in Lemma B.5, B.6
215 and Corollary B.7 (except for $L$ in these Lemmas being replaced by $AL_0$). Other terms in (78) just
216 appears in (77). Proof is completed. $\qquad\qquad\qquad\qquad\qquad\qquad\qquad\qquad\qquad$ $\square$

217 Note that the descent inequality in Lemma B.13 is small in terms of $\|\nabla F(x)\|$ if $\nu$ and $\beta$ are close
218 to 1. We now try to convert the term $\|\tilde{m}\|$ into $\|\nabla F(x)\|$, which is stated in the following lemma.

**Lemma B.14** *Suppose $AL_0\eta \leq c_1(1-\beta)$ and $BL_1\gamma \leq c_3(1-\beta)$ for some constant $c_1$ and $c_3$. Let $\tilde{m}_0 = \nabla F(x_0)$ for simplicity. Let set $\mathcal{S}$ and $\overline{\mathcal{S}}$ be defined in Corollary B.12. Then*

$$
\mathbb{E}\sum_{t\in\overline{\mathcal{S}}}\|\tilde{m}_t\| \geq \frac{1}{1+c_1}\mathbb{E}\left(\sum_{t\in\overline{\mathcal{S}}}(1-c_1(1-\nu\beta)-c_3)\|\nabla F(x_t)\| - c_1\sigma\right)
$$
$$
- \frac{1}{1-\beta}\mathbb{E}\left(\sum_{t\in\mathcal{S}}(AL_0 + BL_1\|\nabla F(x_t)\|)\gamma\right)
$$
(79)

**Proof:** The proof of Lemma B.14 is similar to the proof of Lemma B.8. We first write (44) again as follows:

$$
\sum_{t=0}^{T-1}\|m_t - \nabla F(x_t)\|
$$
$$
\leq \frac{1}{1-\beta}\sum_{t=0}^{T-1}(AL_0 + BL_1\|\nabla F(x_t)\|)\times
$$
$$
\left(\nu\min\left(\eta, \frac{\gamma}{\|m_{t+1}\|}\right)\|m_{t+1}\| + (1-\nu)\min\left(\eta, \frac{\gamma}{\|\nabla f(x_t,\xi_t)\|}\right)\|\nabla f(x_t,\xi_t)\|\right)
$$
$$
\leq \frac{1}{1-\beta}\left(\sum_{t=0}^{T-1}BL_1\gamma\|\nabla F(x_t)\| + \sum_{t\in\mathcal{S}}AL_0\gamma + \sum_{t\in\overline{\mathcal{S}}}AL_0\eta\left((1-\nu)\|\nabla f(x_t,\xi_t)\| + \nu\|m_{t+1}\|\right)\right)
$$
(80)

Therefore,

$$
\sum_{t=0}^{T-1}\|m_t - \nabla F(x_t)\|
$$
$$
\leq \frac{1}{1-\beta}\sum_{t\in\mathcal{S}}(AL_0 + BL_1\|\nabla F(x_t)\|)\gamma
$$
$$
+ \frac{1}{1-\beta}\sum_{t\in\overline{\mathcal{S}}}[AL_0\nu\eta\|\tilde{m}_{t+1}\| + (BL_1\gamma + AL_0(1-\nu)\eta)\|\nabla F(x_t)\| + AL_0\eta\sigma]
$$
$$
\leq \frac{1}{1-\beta}\sum_{t\in\mathcal{S}}(AL_0 + BL_1\|\nabla F(x_t)\|)\gamma +
$$
$$
\frac{1}{1-\beta}\sum_{t\in\overline{\mathcal{S}}}(AL_0\nu\eta\beta\|\tilde{m}_t\| + (AL_0\eta(1-\nu\beta) + BL_1\gamma)\|\nabla F(x_t)\| + AL_0\eta\sigma)
$$
$$
\leq \frac{1}{1-\beta}\sum_{t\in\mathcal{S}}(AL_0 + BL_1\|\nabla F(x_t)\|)\gamma + \sum_{t\in\overline{\mathcal{S}}}((c_1(1-\nu\beta)+c_3)\|\nabla F(x_t)\| + c_1\nu\beta\|m_t\| + c_1\sigma)
$$
(81)

Using $\|\tilde{m}_t\| \geq \|\nabla F(x_t)\| - \|\tilde{m}_t - \nabla F(x_t)\|$ and some straightforward calculation, we obtain

$$
(1+c_1)\sum_{t\in\overline{\mathcal{S}}}\|\tilde{m}_t\| \geq \sum_{t\in\overline{\mathcal{S}}}((1-c_1(1-\nu\beta)-c_3)\|\nabla F(x_t)\| - c_1\sigma)
$$
$$
- \frac{1}{1-\beta}\mathbb{E}\left(\sum_{t\in\mathcal{S}}(AL_0 + BL_1\|\nabla F(x_t)\|)\gamma\right)
$$
(82)

$\square$

We now merge the two cases corresponding to Corollary B.12 and Lemma B.13. The proof of the following theorem involves many techniques which are different from the deterministic case and is far more challenging.

**Theorem B.15** *Let $F^*$ be the optimal value, and $\Delta = F(x_0) - F^*$. Assume $m_0 = \nabla F(x_0)$ for simplicity. Fix $\varepsilon \le 0.1$ be a small constant. If $\gamma \le \frac{\varepsilon}{\sigma} \min\left(\frac{\varepsilon}{AL_0}, \frac{1-\beta}{AL_0}, \frac{1-\beta}{50BL_1}\right)$ and $\gamma/\eta = 5\sigma$ where constants $A = 1.01, B = 1.01$, then*

$$\frac{1}{T} \sum_{t=1}^{T} \mathbb{E}\|\nabla F(x_t)\| \le 2\varepsilon \tag{83}$$

*as long as*

$$T \ge \frac{3}{\varepsilon^2 \eta} \Delta \tag{84}$$

**Proof:** Based on the previous results, we take summation over $t$ and obtain

$$
\sum_{t=0}^{T-1} (G(x_{t+1}, \tilde{m}_{t+1}) - G(x_t, \tilde{m}_t))
$$

$$
\le \frac{12\gamma}{5\beta(1-\beta)}\|\tilde{\delta}_0\| + \left(\frac{12}{5(1-\beta)}AL_0 + \frac{12\gamma}{5\eta(1-\beta)}BL_1 + \frac{1}{2}AL_0\right)\gamma^2 T_{\mathcal{S}}
$$

$$
- \eta \sum_{t=0}^{T-1} (\nu \langle \nabla F(x_t), m_{t+1} - \tilde{m}_{t+1}\rangle + (1-\nu)\langle \nabla F(x_t), \nabla f(x_t, \xi_t) - \nabla F(x_t)\rangle) +
$$

$$
\gamma \sum_{t \in \mathcal{S}} \left[ -\left(\left(\frac{4}{5} \times \frac{2}{5} - \frac{\eta}{\gamma}\sigma\right)\|\nabla F(x_t)\| + \left(\frac{16}{25} \times \frac{3\gamma}{5\eta} - \frac{7}{5}\sigma\right)\right) + \frac{\gamma}{2}BL_1\|\nabla F(x_t)\| + \frac{12\gamma}{5(1-\beta)}BL_1\|\nabla F(x_t)\| \right]
$$

$$
+ \sum_{t \in \overline{\mathcal{S}}} -\frac{\eta}{2}\left((1-\nu\beta)\|\nabla F(x_t)\|^2 + \nu\beta\|\tilde{m}_t\|^2\right) + +\frac{\gamma^2}{2}BL_1\|\nabla F(x)\|
$$

$$
+ \sum_{t \in \overline{\mathcal{S}}} AL_0\eta^2\sigma\|(1-\nu)\nabla F(x_t) + \nu\tilde{m}_{t+1}\| + \frac{AL_0}{2}\eta^2\|(1-\nu)(\nabla f(x_t, \xi_t) - \nabla F(x_t)) + \nu(m_{t+1} - \tilde{m}_{t+1})\|^2
$$

$$\tag{85}$$

We now simplify (85) by taking expectation. We first have

$$\mathbb{E}[\langle \nabla F(x_t), \nabla f(x_t, \xi_t) - \nabla F(x_t)\rangle] = 0 \tag{86}$$

due to the noise assumption. For the term $\mathbb{E}\|(1-\nu)(\nabla f(x_t, \xi_t) - \nabla F(x_t)) + \nu(m_{t+1} - \tilde{m}_{t+1})\|^2$, similarly using the noise assumption and Lemma B.10, we can obtain

$$\mathbb{E}\|(1-\nu)(\nabla f(x_t, \xi_t) - \nabla F(x_t)) + \nu(m_{t+1} - \tilde{m}_{t+1})\|^2 \le \left((1-\beta\nu)^2 + \frac{1-\beta}{1+\beta}\beta^2\nu^2\right)\sigma^2 \tag{87}$$

We now tackle the most challenging part: the expectation of $\langle \nabla F(x_t), m_{t+1} - \tilde{m}_{t+1}\rangle$ for some $t$.

$$
\begin{aligned}
&- \mathbb{E}\langle \nabla F(x_t), m_{t+1} - \tilde{m}_{t+1}\rangle \\
&= -\mathbb{E}[\langle \nabla F(x_t), \beta(m_t - \tilde{m}_t) + (1-\beta)(\nabla f(x_t, \xi_t) - \nabla F(x_t))\rangle] \\
&= -\beta\mathbb{E}\langle \nabla F(x_t), m_t - \tilde{m}_t\rangle \\
&= \beta\mathbb{E}[-\langle \nabla F(x_{t-1}), m_t - \tilde{m}_t\rangle + \langle \nabla F(x_{t-1}) - \nabla F(x_t), m_t - \tilde{m}_t\rangle]
\end{aligned}
\tag{88}
$$

Applying the above equation recursively, we obtain

$$-\mathbb{E}\langle \nabla F(x_t), m_{t+1} - \tilde{m}_{t+1}\rangle \le \mathbb{E}\sum_{\tau=0}^{t-1} \beta^{t-\tau}\langle \nabla F(x_\tau) - \nabla F(x_{\tau+1}), m_{\tau+1} - \tilde{m}_{\tau+1}\rangle \tag{89}$$

Therefore

$$-\mathbb{E}\sum_{t=0}^{T-1}\langle \nabla F(x_t), m_{t+1} - \tilde{m}_{t+1}\rangle \le \frac{\beta}{1-\beta}\sum_{t=0}^{T-1}\max\left(\mathbb{E}\langle \nabla F(x_t) - \nabla F(x_{t+1}), m_{t+1} - \tilde{m}_{t+1}\rangle, 0\right) \tag{90}$$

240 We now bound $\mathbb{E}[\langle \nabla F(x_t) - \nabla F(x_{t+1}), m_{t+1} - \tilde{m}_{t+1} \rangle]$.

$$
\mathbb{E}\left\langle \nabla F(x_t) - \nabla F(x_{t+1}), m_{t+1} - \tilde{m}_{t+1} \right\rangle
$$

$$
= \mathbb{E}\int_0^1 (x_t - x_{t+1})^T \nabla^2 F(\mu x_t + (1-\mu)x_{t+1})(m_{t+1} - \tilde{m}_{t+1})\mathrm{d}\mu
$$

$$
= \mathbb{E}\left[ \min\left(\eta, \frac{\gamma}{\|m_{t+1}\|}\right)\int_0^1 \nu m_{t+1}^T \nabla^2 F(\mu x_t + (1-\mu)x_{t+1})(m_{t+1} - \tilde{m}_{t+1})\mathrm{d}\mu \right]
$$

$$
+ \mathbb{E}\left[ \min\left(\eta, \frac{\gamma}{\|\nabla f(x_t, \xi_t)\|}\right)\int_0^1 (1-\nu)\nabla f(x_t, \xi_t)^T \nabla^2 F(\mu x_t + (1-\mu)x_{t+1})(m_{t+1} - \tilde{m}_{t+1})\mathrm{d}\mu \right]
$$

$$
\leq \mathbb{E}\left[ \min\left(\eta, \frac{\gamma}{\|m_{t+1}\|}\right)\int_0^1 \nu \tilde{m}_{t+1}^T \nabla^2 F(\mu x_t + (1-\mu)x_{t+1})(m_{t+1} - \tilde{m}_{t+1})\mathrm{d}\mu \right]
$$

$$
+ \mathbb{E}\left[ \min\left(\eta, \frac{\gamma}{\|\nabla f(x_t, \xi_t)\|}\right)\int_0^1 (1-\nu)\nabla F(x_t)^T \nabla^2 F(\mu x_t + (1-\mu)x_{t+1})(m_{t+1} - \tilde{m}_{t+1})\mathrm{d}\mu \right]
$$

$$
+ \eta\mathbb{E}[(AL_0 + BL_1\|\nabla F(x_t)\|)]\sigma^2(1-\beta)\left(\frac{\nu}{1+\beta} + 1 - \nu\right)
$$

$$
\leq \mathbb{E}\left[ \min\left(\eta, \frac{\gamma}{\|m_{t+1}\|}\right)\nu(AL_0 + BL_1\|\nabla F(x_t)\|)\|\tilde{m}_{t+1}\|\sigma \right]
$$

$$
+ \mathbb{E}\left[ \min\left(\eta, \frac{\gamma}{\|\nabla f(x_t, \xi_t)\|}\right)(1-\nu)(AL_0 + BL_1\|\nabla F(x_t)\|)\|\nabla F(x_t)\|\sigma \right]
$$

$$
+ \eta\mathbb{E}[(AL_0 + BL_1\|\nabla F(x_t)\|)]\sigma^2(1-\beta)\left(\frac{\nu}{1+\beta} + 1 - \nu\right)
$$

$$
\leq \mathbb{E}\left[\eta(\nu\|\tilde{m}_{t+1}\| + (1-\nu)\|\nabla F(x_t)\|)AL_0\sigma\right]
$$

$$
+ \mathbb{E}\left[\left(\nu\min\left(\eta, \frac{\gamma}{\|m_{t+1}\|}\right)\|\tilde{m}_{t+1}\| + (1-\nu)\min\left(\eta, \frac{\gamma}{\|\nabla f(x_t, \xi_t)\|}\right)\|\nabla F(x_t)\|\right)BL_1\|\nabla F(x_t)\|\sigma\right]
$$

$$
+ \eta\mathbb{E}[(AL_0 + BL_1\|\nabla F(x_t)\|)]\sigma^2(1-\beta)\left(\frac{\nu}{1+\beta} + 1 - \nu\right)
$$

$$
\leq \eta\mathbb{E}\left[(\nu\|\tilde{m}_{t+1}\| + (1-\nu)\|\nabla F(x_t)\|)AL_0\sigma\right] + \eta AL_0\sigma^2(1-\beta)\left(\frac{\nu}{1+\beta} + 1 - \nu\right)
$$

$$
+ \frac{6}{5}\gamma\mathbb{E}\left[BL_1\|\nabla F(x_t)\|\sigma\right] + \frac{1}{5}\gamma\mathbb{E}\left[BL_1\|\nabla F(x_t)\|\sigma\right]
$$

$$
\tag{91}
$$

241 where the first inequality uses the proof of Corollary A.4 and Lemma B.10, and the last inequality
242 uses $\gamma/\eta = 5\sigma$. By taking summation of the above inequality we obtain

$$
-\sum_{t=0}^{T-1}\mathbb{E}\left\langle \nabla F(x_t), m_{t+1} - \tilde{m}_{t+1}\right\rangle \leq \frac{\beta}{1-\beta}\sum_{t=0}^{T-1}\left(\eta AL_0 + \frac{7}{5}\gamma BL_1\right)\sigma\|\nabla F(x_t)\|
$$

$$
+ \eta AL_0\sigma^2\beta\left(\frac{\nu}{1+\beta} + 1 - \nu\right)T + \frac{\nu\beta^2}{(1-\beta)^2}\eta AL_0\sigma\|\nabla F(x_0)\|
$$

$$
\tag{92}
$$

243 where we uses the following inequality to convert $\|\tilde{m}_{t+1}\|$ to $\|\nabla F(x_t)\|$.

$$
\sum_{t=0}^{T-1}\|\tilde{m}_{t+1}\| \leq \frac{\beta}{1-\beta}\|\nabla F(x_0)\| + (1-\beta)\sum_{t=0}^{T-1}\sum_{\tau=0}^{t}\beta^{t-\tau}\|\nabla F(x_\tau)\|
$$

$$
\leq \frac{\beta}{1-\beta}\|\nabla F(x_0)\| + \sum_{t=0}^{T-1}\|\nabla F(x_t)\|
$$

$$
\tag{93}
$$

Combining (85), (86), (87), (92), using inequality (93) to get rid of the term $\|\tilde{m}_t\|$ and applying Lemma B.10, we obtain

$$
\begin{aligned}
&\mathbb{E}\sum_{t=0}^{T-1}(G(x_{t+1},\tilde{m}_{t+1}) - G(x_t,\tilde{m}_t)) \\
&\leq \frac{12\gamma}{5\beta(1-\beta)}\|\tilde{\delta}_0\| + \frac{\nu\beta}{(1-\beta)^2}AL_0\eta^2\sigma\|\nabla F(x_0)\| + \left(\frac{12}{5(1-\beta)}AL_0 + \frac{12\gamma}{5\eta(1-\beta)}BL_1 + \frac{1}{2}AL_0\right)\gamma^2 T_{\mathcal{S}} + \\
&\quad \gamma\mathbb{E}\sum_{t\in\mathcal{S}}\left[-\left(\left(\frac{4}{5}\times\frac{2}{5}-\frac{\eta}{\gamma}\sigma\right)\|\nabla F(x_t)\| + \left(\frac{16}{25}\times\frac{3\gamma}{5\eta}-\frac{7}{5}\sigma\right)\right) + \frac{\gamma}{2}BL_1\|\nabla F(x_t)\| + \frac{12\gamma}{5(1-\beta)}BL_1\|\nabla F(x_t)\|\right] \\
&\quad + \mathbb{E}\sum_{t\in\overline{\mathcal{S}}}\left(-\frac{\eta}{2}(1-\nu\beta)\|\nabla F(x_t)\|^2 - \frac{\eta}{2}\nu\beta\|m_t\|^2 + \frac{\gamma^2}{2}BL_1\|\nabla F(x)\|\right) \\
&\quad + \mathbb{E}\sum_{t=0}^{T-1}\eta^2 AL_0\sigma\left(\|\nabla F(x_t)\| + \left(\frac{(1-\nu\beta)^2}{2} + \frac{1-\beta}{2(1+\beta)}\nu^2\beta^2\right)\sigma\right) \\
&\quad + \frac{\nu\beta\eta\sigma}{1-\beta}\mathbb{E}\left(\sum_{t=0}^{T-1}(AL_0\eta\|\nabla F(x_t)\| + \frac{7}{5}BL_1\gamma\|\nabla F(x_t)\|)\right) + AL_0\eta^2\sigma^2\nu\beta\left(\frac{\nu}{1+\beta}+1-\nu\right)T \\
&= P_0 + \mathbb{E}\left(P_1 T_{\mathcal{S}} + P_2(T - T_{\mathcal{S}}) + \sum_{t\in\mathcal{S}}P_3\|\nabla F(x_t)\| + \sum_{t\in\overline{\mathcal{S}}}P_4\|\nabla F(x_t)\|\right) \\
&\quad - \mathbb{E}\sum_{t\in\overline{\mathcal{S}}}\frac{\eta}{2}\left((1-\beta)\|\nabla F(x_t)\|^2 + \beta\|\tilde{m}_t\|^2\right)
\end{aligned}
\tag{94}
$$

where

$$
\begin{aligned}
P_0 &= \frac{12\gamma}{5\beta(1-\beta)}\|\tilde{\delta}_0\| + \frac{\nu\beta}{(1-\beta)^2}AL_0\eta^2\sigma\|\nabla F(x_0)\| = \frac{\nu\beta}{(1-\beta)^2}AL_0\eta^2\sigma\|\nabla F(x_0)\| \\
P_1 &= -\frac{16}{25}\times\frac{3\gamma^2}{5\eta} + \left(\frac{12\gamma^2}{5(1-\beta)} + \frac{\gamma^2}{2}\right)AL_0 + \frac{12\gamma^3}{5\eta(1-\beta)}BL_1 + \frac{7}{5}\gamma\sigma + P_2 \\
P_2 &= AL_0\eta^2\sigma^2\left(\frac{(1-\nu\beta)^2}{2} + \frac{1-\beta}{2(1+\beta)}\nu^2\beta^2\right) + AL_0\eta^2\sigma^2\nu\beta\left(\frac{\nu}{1+\beta}+1-\nu\right) = \frac{1}{2}\eta^2 AL_0\sigma^2 \\
P_3 &= -\frac{4}{5}\times\frac{2}{5}\gamma + \eta\sigma + \left(\frac{\gamma^2}{2} + \frac{12\gamma^2}{5(1-\beta)} + \frac{\nu\beta\eta\sigma}{1-\beta}\times\frac{7}{5}\gamma\right)BL_1 + \eta^2 AL_0\sigma + \frac{\nu\beta\sigma}{1-\beta}AL_0\eta^2 \\
P_4 &= \eta^2 AL_0\sigma + \frac{\nu\beta\sigma}{1-\beta}\left(AL_0\eta + \frac{7}{5}BL_1\gamma\right)\eta + \frac{\gamma^2}{2}BL_1
\end{aligned}
$$

Let $\gamma \leq \frac{\varepsilon}{2\sigma}\min\left(\frac{\varepsilon}{AL_0}, \frac{1-\beta}{AL_0}, \frac{1-\beta}{25BL_1}\right)$, and fix the ratio $\gamma/\eta = 5\sigma$. Then for small enough $\varepsilon < 0.1$ and large enough noise $\sigma > 1$,

$$
\begin{aligned}
P_1 &\leq \left(-\frac{16}{25}\times 3\sigma + \frac{3\varepsilon}{2\sigma} + \frac{12\varepsilon}{50} + \frac{7}{5}\sigma + \frac{\varepsilon^2}{100\sigma}\right)\gamma \leq -\frac{3}{10}\sigma\gamma \\
P_3 &\leq \left(-\frac{4}{5}\times\frac{2}{5} + \frac{1}{5} + \left(\frac{1-\beta}{2} + \frac{12}{5} + \frac{7}{5}\times\frac{\beta}{5}\right)\frac{\varepsilon}{50\sigma} + \frac{\varepsilon^2}{50\sigma^2} + \frac{\varepsilon}{50\sigma^2}\right)\gamma \leq -\frac{1}{10}\gamma
\end{aligned}
\tag{95}
$$

We can also bound $P_4$ as follows:

$$
\begin{aligned}
P_4 &\leq \frac{1}{1-\beta}AL_0\sigma\eta^2 + \left(\frac{\beta}{1-\beta}\times\frac{7}{5} + \frac{5}{2}\right)BL_1\sigma\gamma\eta \\
&\leq \frac{1}{10}\varepsilon\eta + \left(\frac{\beta}{1-\beta}\times\frac{7}{5} + \frac{5}{2}\right)\frac{\varepsilon}{50}(1-\beta)\eta \\
&\leq \frac{1}{10}\varepsilon\eta + \frac{1}{20}\varepsilon\eta = \frac{3}{20}\varepsilon\eta
\end{aligned}
\tag{96}
$$

250 Applying the above estimates and rearranging (94), we have

$$G(x_0) - F^* + P_0$$

$$\geq \mathbb{E}\left[\sum_{t\in\mathcal{S}}\left(\frac{3}{10}\sigma\gamma + \frac{1}{10}\gamma\|\nabla F(x_t)\|\right) + \sum_{t\in\overline{\mathcal{S}}}\left(\frac{\eta}{2}\left((1-\nu\beta)\|\nabla F(x_t)\|^2 + \nu\beta\|m_t\|^2\right) - \frac{AL_0}{2}\sigma^2\eta^2 - \frac{3}{20}\varepsilon\eta\|\nabla F(x_t)\|\right)\right]$$

$$\geq \mathbb{E}\left[\sum_{t\in\mathcal{S}}\left(\frac{3}{10}\sigma\gamma + \frac{1}{10}\gamma\|\nabla F(x_t)\|\right) + \sum_{t\in\overline{\mathcal{S}}}\left(\frac{\eta}{2}\left((1-\nu\beta)\|\nabla F(x_t)\|^2\right) - \frac{AL_0}{2}\sigma^2\eta^2 - \frac{3}{20}\varepsilon\eta\|\nabla F(x_t)\|\right)\right]$$

$$+ \frac{1}{2}\eta\nu\beta\mathbb{E}\left[\sum_{t\in\overline{\mathcal{S}}}\left(2\varepsilon\|\tilde{m}_t\| - \varepsilon^2\right)\right]$$

$$(97)$$

251 Due to Lemma B.14 ($AL_0\eta\sigma \leq \frac{\varepsilon}{10}(1-\beta), BL_1\gamma \leq \frac{\varepsilon}{50}(1-\beta)$), we clearly have

$$\mathbb{E}\sum_{t\in\overline{\mathcal{S}}}\|\tilde{m}_t\|$$

$$\geq \left(1 - \frac{\varepsilon}{10}\right)\mathbb{E}\left[\sum_{t\in\overline{\mathcal{S}}}\left(\left(1 - \frac{\varepsilon}{5}\right)\|\nabla F(x_t)\| - \frac{\varepsilon}{10}\right)\right] - \mathbb{E}\left[\sum_{\tau\in S}\left(\frac{\gamma}{1-\beta}(AL_0 + BL_1\|\nabla F(x_\tau)\|)\right)\right]$$

$$\geq \left(1 - \frac{3}{10}\varepsilon\right)\mathbb{E}\left[\sum_{t\in\overline{\mathcal{S}}}(\|\nabla F(x_t)\|)\right] - \frac{\varepsilon}{10}(T - T_{\mathcal{S}}) - \mathbb{E}\left[\sum_{\tau\in S}\left(\frac{\gamma}{1-\beta}(AL_0 + BL_1\|\nabla F(x_\tau)\|)\right)\right]$$

$$(98)$$

252 Define

$$U(x) := \left(\frac{1}{10}\gamma - \frac{\nu\beta}{1-\beta}BL_1\varepsilon\gamma\eta\right)\|\nabla F(x)\| + \left(\frac{3}{10}\sigma\gamma - \frac{\nu\beta}{1-\beta}AL_0\varepsilon\gamma\eta\right)$$

$$V(x) := \frac{1}{2}\eta(1-\nu\beta)\|\nabla F(x)\|^2 + \left(\frac{19}{20}\nu\beta\eta\varepsilon - \frac{3}{20}\varepsilon\eta\right)\|\nabla F(x)\| - \left(\frac{1}{2}AL_0\sigma^2\eta^2 + \frac{1}{2}\nu\beta\varepsilon^2\eta + \frac{1}{10}\nu\beta\varepsilon^2\eta\right)$$

$$(99)$$

253 Plugging (98) into (97), we obtain

$$G(x_0) - F^* + P_0 \geq \mathbb{E}\left[\sum_{t\in\mathcal{S}}U(x_t) + \sum_{t\in\overline{\mathcal{S}}}V(x_t)\right]$$

$$= \mathbb{E}\left[\sum_{t=1}^{T}\left(\mathbb{I}_{t\in\mathcal{S}}U(x_t) + \mathbb{I}_{t\in\overline{\mathcal{S}}}V(x_t)\right)\right] \geq \mathbb{E}\left[\sum_{t=0}^{T-1}\min\{U(x_t), V(x_t)\}\right]$$

$$(100)$$

254 Since

$$U(x) \geq \left(\frac{1}{10} - \frac{\nu\beta\varepsilon^2}{50\sigma^2}\right)\gamma\|\nabla F(x)\| + \left(\frac{3}{10}\sigma\gamma - \frac{1}{10\sigma^2}\nu\beta\varepsilon^2\gamma\right)$$

$$\geq \frac{1}{20}\gamma\|\nabla F(x)\| + \frac{1}{5}\sigma\gamma$$

$$(101)$$

255

$$V(x) \geq \frac{1}{2}\eta(1-\nu\beta)\|\nabla F(x)\|^2 + \left(\frac{19}{20}\nu\beta\eta\varepsilon - \frac{3}{20}\varepsilon\eta\right)\|\nabla F(x)\| - \left(\frac{1}{20} + \frac{3}{5}\nu\beta\right)\varepsilon^2\eta$$

$$\geq \frac{1}{2}\eta(1-\nu\beta)\left(2\varepsilon\|\nabla F(x)\| - \varepsilon^2\right) + \left(\frac{19}{20}\nu\beta\eta\varepsilon - \frac{3}{20}\varepsilon\eta\right)\|\nabla F(x)\| - \left(\frac{1}{20} + \frac{3}{5}\nu\beta\right)\varepsilon^2\eta$$

$$\geq \frac{4}{5}\varepsilon\eta\|\nabla F(x)\| - \frac{4}{5}\varepsilon^2\eta$$

$$(102)$$

It clearly follows that $\min\{U(x), V(x)\} \geq \frac{4}{5}\varepsilon\eta\|\nabla F(x)\| - \frac{4}{5}\varepsilon^2\eta$. Therefore

$$G(x_0) - F^* + P_0 \geq \frac{4}{5}\varepsilon\eta\mathbb{E}\sum_{t=0}^{T-1}(\|\nabla F(x)\| - \varepsilon) \tag{103}$$

Therefore, as long as $T > \frac{5}{4\varepsilon^2\eta}\left(G(x_0) - F^* + P_0\right)$, we have $\frac{1}{T}\mathbb{E}\left[\sum_{t=1}^{T}\|\nabla F(x_t)\|\right] < 2\varepsilon$.

We finally show $G(x_0) - F^* + P_0 = O(F(x_0) - F^*)$. Using Lemma A.5,

$$\frac{1}{1-\beta}\min\left(\gamma\|m_0\|, \eta\|m_0\|^2\right) \leq \frac{1}{50}\min\left(\frac{\|\nabla F(x_0)\|}{L_1}, \frac{\|\nabla F(x_0)\|^2}{L_0}\right) \leq \frac{8}{50}(F(x_0) - F^*) \tag{104}$$

For the term $P_0$, if $\|\nabla F(x_0)\| = \Omega(L_0/L_1)$, we can similarly use Lemma A.5 to obtain $P_0 = \mathcal{O}(F(x_0) - F^*)$. If $\|\nabla F(x_0)\| = \mathcal{O}(L_0/L_1)$, using $L_0\|\nabla F(x_0)\| \leq L_1\|\nabla F(x_0)\|^2$ and Lemma A.5 leads to the result. □

## Appendix C   Discussion of the normalized momentum algorithm

In this section we analyze in detail the theoretical aspects of the normalized momentum algorithm, as well as some practical issues. Recall that this algorithm can be seen as a special case of our clipping framework. For convenience we re-write it in Algorithm 2.

---
**Algorithm 2:** The Stochastic Normalized Momentum Algorithm(SNM)

---
**Input :** Initial point $x_0$, initial momentum $m_0$, the learning rate $\eta$, momentum factor $\beta$ and the total number of iterations $T$

1 **for** $i \leftarrow 1$ **to** $T$ **do**
2     $m_t \leftarrow \beta m_{t-1} + (1-\beta)\nabla f(x_{t-1}, \xi_{t-1})$;
3     $x_t \leftarrow x_{t-1} - \eta\dfrac{m_t}{\|m_t\|}$;

---

We remark that SNM is different from the clipping methods in traditional sense, in that it makes a *normalized* update each iteration. This algorithm has been analyzed in Cutkosky and Mehta [2020] for $L$-smooth functions. In that setting they were able to prove that SNM achieves a complexity of $\mathcal{O}(\Delta L\sigma^2\varepsilon^{-4})$.

For $(L_0, L_1)$-smooth functions, we show that: *(a).* With carefully chosen momentum parameter $\beta$ and step size $\eta$, SNM can achieve a complexity of $\mathcal{O}(\Delta L_0\sigma^2\varepsilon^{-4})$, which is the same as the complexity we obtain in Theorem 3.2. *(b).* There are some practical issues that make SNM less favorable than traditional clipping methods (such as the other three special cases of our framework discussed in Section 3 of the main paper).

The following results provides convergence guarantee for Algorithm 2.

**Lemma C.1** *Consider the algorithm that starts at $x_0$ and make updates $x_{t+1} = x_t - \eta m_{t+1}$. Define $\delta_t := m_{t+1} - \nabla F(x_t)$ be the estimation error. Assume $\eta \leq c/L_1$ for some $c > 0$ and let constants $A = 1 + e^c - \frac{e^c-1}{c}, B = \frac{e^c-1}{c}$. Then*

$$F(x_{t+1}) - F(x_t) \leq -\left(\eta - \frac{1}{2}BL_1\eta^2\right)\|\nabla F(x_t)\| + \frac{1}{2}AL_0\eta^2 + 2\eta\|\delta_t\|$$

*And thus, by a telescope sum we have*

$$\left(1 - \frac{1}{2}BL_1\eta\right)\sum_{t=0}^{T-1}\|\nabla F(x_t)\| \leq \frac{F(x_0) - F(x_T)}{\eta} + \frac{1}{2}AL_0T\eta + 2\sum_{t=0}^{T-1}\|\delta\|$$

**Proof:** Since $\|x_{t+1} - x_t\| = \eta_t$, by Lemma A.3 we have

$$F(x_{t+1}) - F(x_t) \leq -\frac{\eta}{\|m_{t+1}\|} \langle \nabla F(x_t), m_{t+1}\rangle + \frac{1}{2}\eta^2 (AL_0 + BL_1\|\nabla F(x_t)\|)$$

$$\leq \eta\left(-\|\nabla F(x_t)\| + 2\|\delta_t\|\right) + \frac{1}{2}\eta^2 (AL_0 + BL_1\|\nabla F(x_t)\|)$$

$$\leq -\left(\eta - \frac{1}{2}BL_1\eta^2\right)\|\nabla F(x_t)\| + \frac{1}{2}AL_0\eta^2 + 2\eta\|\delta_t\|$$

where in the second inequality we use Lemma B.1. $\qquad\square$

**Theorem C.2** *Suppose that Assumptions 1,2 and 4 holds, and $\Delta = F(x_0) - F^*$ where $F^* = \inf_{x\in\mathbb{R}^d} F(x)$. Let $m_0 = \nabla F(x_0)$ in Algorithm 2 for simplicity, and denote $\alpha = 1 - \beta$. If we choose $\eta = \Theta\left(\min(L_1^{-1}, L_0^{-1}\varepsilon)\alpha\right)$ and $\alpha = \Theta\left(\sigma^{-2}\varepsilon^2\right)$, then as long as $\varepsilon = \mathcal{O}\left(\min\left\{\frac{L_0}{L_1}, \sigma\right\}\right)$, we have*

$$\frac{1}{T}\sum_{t=0}^{T-1}\mathbb{E}\left[\|\nabla F(x_t)\|\right] \leq \varepsilon$$

*holds in $T = \mathcal{O}(\Delta L_0\sigma^2\varepsilon^{-4})$ iterations.*

**Proof:** Define the estimation errors $\delta_t := m_{t+1} - \nabla F(x_t)$. Denote $S(a,b) := \nabla F(a) - \nabla F(b)$, then for $a, b$ such that $\|a - b\| = \eta \leq c/L_1$, we can upper bound $S(a,b)$ using Corollary A.4:

$$\|S(a,b)\| \leq \eta\left(AL_0 + BL_1\|\nabla F(b)\|\right) \tag{105}$$

We can use $S(a,b)$ to get a recursive relationship:

$$\delta_{t+1} = \beta m_{t+1} + (1-\beta)\nabla f(x_{t+1}, \xi_{t+1}) - \nabla F(x_{t+1})$$
$$= \beta S(x_t, x_{t+1}) + \beta\delta_t + (1-\beta)(\nabla f(x_{t+1}, \xi_{t+1}) - \nabla F(x_{t+1})) \tag{106}$$

Denote $\delta_t' = \nabla f(x_t, \xi_t) - \nabla F(x_t)$, then

$$\delta_t = \beta\sum_{\tau=0}^{t-1}\beta^\tau S(x_{t-\tau-1}, x_{t-\tau}) + (1-\beta)\sum_{\tau=0}^{t-1}\beta^\tau \delta_{t-\tau}' + (1-\beta)\beta^t\delta_0'$$

Using triangle inequality and plugging in the estimate (105) , we have

$$\|\delta_t\| \leq (1-\beta)\left\|\sum_{\tau=0}^{t}\beta^\tau\delta_{t-\tau}'\right\| + \beta\eta\sum_{\tau=0}^{t-1}\beta^\tau\left(AL_0 + BL_1\|\nabla F(x_{t-\tau-1})\|\right) \tag{107}$$

Taking a telescope summation of 107 and using Assumption 2.4 we obtain

$$\mathbb{E}\left[\sum_{t=0}^{T-1}\|\delta_t\|\right] \leq T(1-\beta)\sqrt{\sum_{\tau=0}^{+\infty}\beta^{2\tau}\sigma^2} + \frac{AL_0\eta T}{1-\beta} + \frac{BL_1\eta}{1-\beta}\sum_{t=0}^{T-1}\mathbb{E}\left[\|\nabla F(x_t)\|\right]$$

$$\leq \sqrt{\alpha}T\sigma + \frac{ATL_0\eta}{\alpha} + \frac{BL_1\eta}{\alpha}\sum_{t=0}^{T-1}\mathbb{E}\left[\|\nabla F(x_t)\|\right] \tag{108}$$

Now we use Lemma C.1:

$$\left(1 - \left(\frac{1}{2} + \frac{2}{\alpha}\right)BL_1\eta\right)\mathbb{E}\sum_{t=0}^{T-1}\|\nabla F(x_t)\| \leq \frac{\Delta}{\eta} + \frac{1}{2}AL_0T\eta + 2\left(\sqrt{\alpha}T\sigma + \frac{AL_0\eta T}{\alpha}\right)$$

If we choose $\eta = \Theta\left(\min(L_1^{-1}, L_0^{-1}\varepsilon)\alpha\right)$ and $\alpha = \Theta\left(\sigma^{-2}\varepsilon^2\right)$, then

$$\left(1 - \left(\frac{1}{2} + \frac{2}{\alpha}\right)BL_1\eta\right) = \Theta(1)$$

In this case

$$\frac{1}{T}\mathbb{E}\sum_{t=0}^{T-1}\|\nabla F(x_t)\| = \mathcal{O}\left(\frac{\Delta}{\eta T} + \frac{1}{2}AL_0\eta + \sqrt{\alpha}\sigma + \frac{AL_0\eta}{\alpha}\right) = \mathcal{O}\left(\frac{\Delta}{\eta T} + \varepsilon\right)$$

284   Therefore for $T = \Theta\left(\frac{\Delta}{\eta\varepsilon}\right)$, we have $\frac{1}{T}\mathbb{E}\sum_{t=0}^{T-1}\|\nabla F(x_t)\| = \mathcal{O}(\varepsilon)$. If $\varepsilon = \mathcal{O}(L_0/L_1)$, then $\frac{\Delta}{\eta\varepsilon}$
285   reduces to $\Delta L_0\sigma^2\varepsilon^{-4}$.                            □

286   We have shown the theoretical superiority of Algorithm 2. Specifically, it enjoys the same complex-
287   ity as Theorem 3.2. However we notice some potential drawbacks of Algorithm 2:

-  *Firstly,* the step size of Algorithm 2 is at the order of $\mathcal{O}\left(\varepsilon^3\right)$, while the step size we chose in Theorem 3.2 is $\mathcal{O}\left(\varepsilon^2\right)$. Previous works have noticed that a smaller step size makes it easier to be trapped in a sharp local minima , which may result in worse generalization [Kleinberg et al., 2018].

-  *Secondly,* although the complexity of Algorithm 2 is the same as Theorem 3.2 for small $\varepsilon$, it requires a more restrictive upper bound of $\varepsilon$ to ensure the $\varepsilon^{-4}$ term dominates. For instance with a poor initialization, $\Delta$ may very large. This suggests that in practice, where we do not get into a very small neighbourhood of stationary point, the performance of Algorithm 2 may be worse.

# Appendix D   Details of Lower Bounds in Section 3.3

298   In this section we discuss the lower bound for SGD in Drori and Shamir [2019] in detail. The
299   following result is taken from this paper:

300   **Theorem D.1 [Theorem 2 in Drori and Shamir [2019]]** *Consider a first-order method that given*
301   *a function $F : \mathbb{R}^d \to \mathbb{R}$ and an initial point $x_0 \in \mathbb{R}^d$ generates a sequence of points $\{x_i\}$ satisfying*

$$x_{t+1} = x_t + \eta_{x_0,\ldots,x_t} \cdot (\nabla F(x_t) + \xi_t), \quad t \in [T-1]$$

302   *where $\xi_i$ are some random noise vectors, and returns a point $x_{out} \in \mathbb{R}^d$ as a non-negative linear*
303   *combination of the iterates:*

$$x_{out} = \sum_{t=0}^{T} \zeta_{x_0,\ldots,x_T}^{(t)} x_t$$

304   *We further assume that the step sizes $\eta_{x_0,\ldots,x_t}$ and aggregation coefficients $\zeta_{x_0,\ldots,x_T}^{(t)}$ are deter-*
305   *ministic functions of the norms and inner products between the vectors $x_0,\ldots,x_t, \nabla F(x_0) +$*
306   *$\xi_0,\ldots,\nabla F(x_t) + \xi_t$. Then for any $L, \Delta, \sigma > 0$ and $T \in \mathbb{N}$ there exists a function $F : \mathbb{R}^d \mapsto \mathbb{R}$*
307   *with $L-$Lipschitz gradient, a point $x_0 \in \mathbb{R}^d$ and independent random variables $\xi_t$ with $\mathbb{E}[\xi_t] = 0$*
308   *and $\mathbb{E}\left[\|\xi_t\|^2\right] = \sigma^2$ such that $\forall t \in [T]$*

$$F(x_0) - F(x_t) \overset{a.s.}{\leq} \Delta$$
$$\nabla F(x_t) \overset{a.s.}{=} \gamma$$

309   *and in addition*

$$F(x_0) - F(x_{out}) \overset{a.s.}{\leq} \Delta + \frac{\sigma}{2L}\sqrt{\frac{L\Delta}{T}}$$
$$\nabla F(x_{out}) \overset{a.s.}{=} \gamma$$

311   *where $\gamma \in \mathbb{R}^d$ is a vector such that*

$$\|\gamma\|^2 = \frac{\sigma}{2}\sqrt{\frac{L\Delta}{T}}$$

312   Now we discuss why this shows the optimality of clipped SGD under Assumptions 2.1, 2.2 and 2.4.

313   *Firstly*, Theorem D.1 assumes an upper bound $\Delta$ on $F(x_0) - F(x_t)$ rather than the one assumed
314   in Assumption 2.1 ($F(x_0) - F^* \leq \Delta$). However, in fact we only need to assume that $F(x_0) -$
315   $F(x_T) \leq \Delta$ to prove Theorem 3.2 for clipped SGD. The reason is as follows. In fact, since $\beta =$
316   $0$ for clipped SGD , the momentum term in the energy function disappears, as well as the term
317   $\frac{\nu\beta}{(1-\beta)^2}AL_0\eta^2\sigma\|\nabla F(x_0)\|$ in (94). So we no longer need to use Lemma A.5 to bound the term
318   $\|\nabla F(x_0)\|$. The rest of the proof only needs $F(x_0) - F(x_T) \leq \Delta$ (which is used in the telescope
319   sum in (94)).

320 *Secondly*, although Theorem D.1 only assume that the variance of stochastic gradient is bounded, in
321 their construction the noise is actually defined as

$$P\left(\boldsymbol{\xi}_t = \pm\sigma\mathbf{e}_{t+1}\right) = \frac{1}{2}, \quad t \in [T-1] \tag{109}$$

322 Therefore the norm of the noise is bounded by $\sigma$, and the example used to prove Theorem D.1 still
323 works under Assumption 2.4.

324 Now suppose we need an output such that $\|\nabla f(x_{\text{out}})\| = \|\gamma\| \leq \varepsilon$, then it follows from Theorem
325 D.1 that $T = \Omega\left(L\Delta\sigma^2\varepsilon^{-4}\right)$. Therefore we have shown the optimality of clipped SGD in this class
326 of algorithms, as stated in Section 3.3.

## Appendix E  Justifications on the Mixed Clipping

328 We will show in this section that combining gradient and momentum can be better than using only
329 one of them. We consider a basic optimization problem: $\min_{x\in\mathbb{R}} F(x) = \min_{x\in\mathbb{R}} \mathbb{E}_\xi[f(x,\xi)]$
330 where $f(x,\xi) = \frac{1}{2}(x+\xi)^2$, and the noise $\xi \in \mathbb{R}$ follows the uniform distribution $U[-\sqrt{3}, \sqrt{3}]$ so
331 that $\mathbb{E}[\xi^2] = 1$. To simplify the analysis, we set $\gamma$ in Algorithm 1 to be sufficiently large such that
332 clipping will never be triggered, since the function $F(x) = \frac{1}{2}x^2$ is (1,0)-smooth.

333 In the above optimization problem, the general update formula can be written as:

$$\begin{aligned} m_{t+1} &= \beta m_t + (1-\beta)(x_t + \xi_t) \\ x_{t+1} &= x_t - \nu\eta m_{t+1} - (1-\nu)\eta(x_t + \xi_t) \end{aligned} \tag{110}$$

334 We have the following proposition:

335 **Proposition E.1** *Let $x_0, m_0 \in \mathbb{R}$ be arbitrary real numbers. Let $\xi_i$s be i.i.d. random noises such*
336 *that $\mathbb{E}[\xi_i^2] = 1$. Let the sequence $\{x_t\}$ be defined in (110), where $0 < \eta < 1, 0 \leq \beta < 1$ and*
337 *$0 \leq \nu \leq 1$ are constant hyper-parameters. Then in the limit*

$$\lim_{t\to\infty} \mathbb{E}[F(x_t)] = \frac{\eta}{2} \times \frac{(1+\beta)(1-\beta+\beta\eta) - \nu\eta\beta(1+3\beta-2\nu\beta)}{(2-\eta)(1+\beta)(1-\beta+\beta\eta) - \nu\eta\beta(4\beta-\eta-3\beta\eta+2\nu\eta\beta)} \tag{111}$$

338 We now analyze three cases based on the proposition:

- Only use gradient in an update. Set $\nu = 0$ in (111), we obtain $\lim_{t\to\infty} \mathbb{E}[F(x_t)] = \frac{\eta}{4-2\eta}$.

- Only use momentum in an update. Set $\nu = 1$ in (111), we obtain $\lim_{t\to\infty} \mathbb{E}[F(x_t)] = \frac{\eta}{4-2\eta\frac{1-\beta}{1+\beta}}$.

- Combine gradient and momentum in an update. It can be verified that for proper $0 < \nu < 1$, (111) is less than $\frac{\eta}{4-2\eta\frac{1-\beta}{1+\beta}}$ (therefore less than $\frac{\eta}{4-2\eta}$). Furthermore, when $\beta \to 1$, a straightforward calculation shows that $\lim_{t\to\infty} \mathbb{E}[F(x_t)] \to \frac{\eta}{\frac{4}{1-\nu}-2\eta}$. Thus $\lim_{t\to\infty} \mathbb{E}[F(x_t)]$ can be arbitrarily close to zero if $\nu$ is close to 1. However, this does not happen in the previous two cases, where $\lim_{t\to\infty} \mathbb{E}[F(x_t)]$ there must be greater than $\frac{\eta}{4}$.

347 We further plot the value of (111) with respect to $\nu$ and $\beta$ in Figure 1 to visualize the above finding.
348 It can be clearly seen that the using both gradient and momentum with a proper interpolation factor
349 $\nu$ outperforms both SGD and SGD with momentum by a large margin (Figure 1(a)). Furthermore,
350 we can drive $\beta \to 1$ to further improve convergence (Figure 1(b)), while in SGD with momentum
351 we can not.

352 Although we use the simple function $F(x) = \frac{1}{2}x^2$ as an example, similar result exists in any gen-
353 eral quadratic form with positive definite Hessian. Furthermore, the experiments in Section 4 also
354 demonstrate that the mixed clipping outperforms both gradient clipping and momentum clipping.

Figure 1: Convergence value of different hyper-parameters $\eta, \beta, \nu$ over stochastic function $f(x, \xi) = \frac{1}{2}(x + \xi)^2$. The mixed update with proper $\nu$ outperforms both SGD and SGD with momentum by a large margin. Furthermore, for the mixed update we can drive $\beta \to 1$ to further improve convergence, while for SGD with momentum we can not.

## E.1 Proof of Proposition E.1

### E.1.1 Proof of a simple case

For clarity, we first assume $\nu = 1$. Consider a specific time step $t$. We first calculate $\mathbb{E}[m_t^2]$.

$$
\begin{aligned}
\mathbb{E}[m_{t+1}^2] &= \mathbb{E}[(\beta m_t + (1 - \beta)(x_t + \xi_t))^2] \\
&= \beta^2 \mathbb{E}[m_t^2] + (1 - \beta)^2 (\mathbb{E}[x_t^2] + \mathbb{E}[\xi_t^2]) + 2(1 - \beta)^2 \mathbb{E}[x_t \xi_t] + 2\beta(1 - \beta)(\mathbb{E}[m_t x_t] + \mathbb{E}[m_t \xi_t]) \\
&= \beta^2 \mathbb{E}[m_t^2] + (1 - \beta)^2 (\mathbb{E}[x_t^2] + 1) + 2\beta(1 - \beta)\mathbb{E}[m_t x_t])
\end{aligned}
\tag{112}
$$

where we use the fact that $\xi_t$ is independent with $x_t$ and $m_t$. We then calculate $\mathbb{E}[x_t^2]$.

$$
\begin{aligned}
\mathbb{E}[x_{t+1}^2] &= \mathbb{E}[(x_t - \eta m_t)^2] \\
&= \mathbb{E}[x_t^2 + \eta^2 m_{t+1}^2 - 2\eta x_t m_{t+1}] \\
&= (1 + \eta^2(1 - \beta)^2 - 2\eta(1 - \beta))\mathbb{E}[x_t^2] + \eta^2 \beta^2 \mathbb{E}[m_t^2] + \eta^2(1 - \beta)^2 + 2(\eta^2 \beta(1 - \beta) - \eta\beta)\mathbb{E}[m_t x_t]
\end{aligned}
\tag{113}
$$

where in the last equation we use (112). To complete the recursive relationship, we also need to calculate $\mathbb{E}[x_{t+1} m_{t+1}]$.

$$
\begin{aligned}
\mathbb{E}[x_{t+1} m_{t+1}] &= \mathbb{E}[m_{t+1} x_t - \eta m_{t+1}^2] \\
&= \mathbb{E}[\beta m_t x_t + (1 - \beta)x_t^2 - \eta m_{t+1}^2] \\
&= (1 - \beta)(1 - \eta(1 - \beta))\mathbb{E}[x_t^2] - \eta\beta^2 \mathbb{E}[m_t^2] - \eta(1 - \beta)^2 + (\beta - 2\eta\beta(1 - \beta))\mathbb{E}[m_t x_t]
\end{aligned}
\tag{114}
$$

Combining (112), (113) and (114), we can write the recursive relationship into a matrix form:

$$
\begin{pmatrix} \mathbb{E}x_{t+1}^2 \\ \mathbb{E}m_{t+1}^2 \\ \mathbb{E}x_{t+1}m_{t+1} \\ 1 \end{pmatrix} = \begin{pmatrix} 1 + \eta^2(1 - \beta)^2 - 2\eta(1 - \beta) & \eta^2\beta^2 & 2(\eta^2\beta(1 - \beta) - \eta\beta) & \eta^2(1 - \beta)^2 \\ (1 - \beta)^2 & \beta^2 & 2\beta(1 - \beta) & (1 - \beta)^2 \\ (1 - \beta)(1 - \eta(1 - \beta)) & -\eta\beta^2 & \beta - 2\eta\beta(1 - \beta) & -\eta(1 - \beta)^2 \\ 0 & 0 & 0 & 1 \end{pmatrix} \begin{pmatrix} \mathbb{E}x_t^2 \\ \mathbb{E}m_t^2 \\ \mathbb{E}x_t m_t \\ 1 \end{pmatrix}
\tag{115}
$$

Denote the above matrix as $M$. After a straightforward calculation, we can find that $\lambda_1 = 1$ is an eigenvalue of $M$, and

$$
u = \left( -\eta \frac{1 + \beta}{1 - \beta}, -2, \eta, \eta - 2\frac{1 + \beta}{1 - \beta} \right)^T
$$

is the only eigenvector associated with $\lambda_1 = 1$. Similarly, $\lambda_2 = \beta$ is also an eigenvalue of $M$. Let the other two eigenvalues be $\lambda_3$ and $\lambda_4$, then

$$
\lambda_1 \lambda_2 \lambda_3 \lambda_4 = \det M = \beta^3
$$

$$
\lambda_1 + \lambda_2 + \lambda_3 + \lambda_4 = \operatorname{tr} M = 1 - \beta + (\eta(1 - \beta) - \beta - 1)^2
$$

It follows that $\lambda_3\lambda_4 = \beta^2$ and $\lambda_3 + \lambda_4 = (\eta(1-\beta) - \beta - 1)^2 - 2\beta$. Since $(1 + \beta - \eta(1-\beta))^2 <$ $(1 + \beta)^2$, we have $\lambda_3 + \lambda_4 < 1 + \beta^2$. Therefore $|\lambda_3| < 1$ and $|\lambda_4| < 1$ (note that $\lambda_3$ and $\lambda_4$ can be composite numbers). If $\eta < 1$, we can further conclude that the four eigenvalues are different from each other (otherwise $\lambda_3 = \lambda_4 = \beta$, which contradicts to $\lambda_3 + \lambda_4 = (\eta(1-\beta) - \beta - 1)^2 - 2\beta$).

Based on the above calculation, for any initial vector $v$, $\lim_{t\to\infty} M^t v$ converges to a vector proportional to $u$. In our case, $(\mathbb{E}x_0^2, \mathbb{E}m_0^2, \mathbb{E}x_0 m_0, 1)^T = (0, 0, 0, 1)^T$, and we also know that the the last element of the vector $\lim_{t\to\infty} M^t(0, 0, 0, 1)^T$ is 1. As a result,

$$\lim_{t\to\infty} M^t(0, 0, 0, 1)^T = -\frac{1 - \beta}{2(1 + \beta)}u$$

Namely,

$$\lim_{t\to\infty} \mathbb{E}[x_{t+1}^2] = \frac{\eta}{2 - \eta\frac{1-\beta}{1+\beta}}$$

### E.1.2   Proof of the general case

Now we prove Proposition E.1 for general $\nu$.

$$\mathbb{E}[x_{t+1}^2] = (1 - \eta + \nu\eta\beta)^2\mathbb{E}[x_t^2] + \nu^2\eta^2\beta^2\mathbb{E}[m_t^2] - 2\nu\eta\beta(1 + \nu\eta\beta - \eta)\mathbb{E}[m_t x_t] + (\nu\eta(1-\beta) + (1-\nu)\eta)^2 \tag{116}$$

$$\mathbb{E}[x_{t+1}m_{t+1}] = (1 - \eta + \nu\eta\beta)(1 - \beta)\mathbb{E}[x_t^2] - \nu\eta\beta^2\mathbb{E}[m_t^2] + (1 - \eta - \nu\eta + 2\nu\eta\beta)\beta E[x_t m_t]$$
$$- \nu\eta(1-\beta)^2 - (1-\nu)\eta(1-\beta) \tag{117}$$

Combining (112), (116) and (117), we obtain the following recursive matrix $M$:

$$M = \begin{pmatrix} (1 - \eta + \nu\eta\beta)^2 & \nu^2\eta^2\beta^2 & -2\nu\eta\beta(1 + \nu\eta\beta - \eta) & (\nu\eta(1-\beta) + (1-\nu)\eta)^2 \\ (1-\beta)^2 & \beta^2 & 2\beta(1-\beta) & (1-\beta)^2 \\ (1 - \eta + \nu\eta\beta)(1-\beta) & -\nu\eta\beta^2 & (1 - \eta - \nu\eta + 2\nu\eta\beta)\beta & -\nu\eta(1-\beta)^2 - (1-\nu)\eta(1-\beta) \\ 0 & 0 & 0 & 1 \end{pmatrix}$$

Using the same calculation as in the previous section, we finally get

$$\lim_{t\to\infty} \mathbb{E}[x_{t+1}^2] = \eta\frac{(1 + \beta)(1 - \beta + \beta\eta) - \nu\eta\beta(1 + 3\beta - 2\nu\beta)}{(2 - \eta)(1 + \beta)(1 - \beta + \beta\eta) - \nu\eta\beta(4\beta - \eta - 3\beta\eta + 2\nu\eta\beta)} \tag{118}$$

## Appendix F   Soft Clipping

---

**Algorithm 3:** The General Soft Clipping Framework

---

**Input :** Initial point $x_0$, learning rate $\eta$, clipping parameter $\gamma$, momentum $\beta \in [0, 1)$, interpolation parameter $\nu \in [0, 1]$ and the total number of iterations $T$

1 Initialize $m_0$ arbitrarily;
2 **for** $t \leftarrow 0$ *to* $T - 1$ **do**
3 $\quad$ Compute the stochastic gradient $\nabla f(x_t, \xi_t)$ for the current point $x_t$;
4 $\quad m_{t+1} \leftarrow \beta m_t + (1 - \beta)\nabla f(x_t, \xi_t)$;
5 $\quad x_{t+1} \leftarrow x_t - \left[\nu\eta\frac{m_{t+1}}{1 + \eta\|m_{t+1}\|/\gamma} + (1 - \nu)\eta\frac{\nabla f(x_t, \xi_t)}{1 + \eta\|\nabla f(x_t, \xi_t)\|/\gamma}\right]$;

---

For Algorithm 1, as long as the norm of the gradient (or momentum) exceeds a constant, it is then clipped; we refer to this form of clipping as *hard clipping*. One can also consider a *soft* form of clipping, as presented in Algorithm 3.

We take $\nu = 0$ for example to analyze soft clipping. For any gradient norm $l_g$, the norm of the update $l_u$ is a function of $l_g$:

$$l_u = h_{\text{soft}}(l_g) = \eta\frac{l_g}{1 + \eta l_g/\gamma} \tag{119}$$

Figure 2: The update norm $l_\mathrm{u}$ w.r.t. the gradient norm $l_\mathrm{g}$ for hard clipping and soft clipping ($\eta = 1, \gamma = 1$).

For hard clipping, we can similarly write

$$l_\mathrm{u} = h_\mathrm{hard}(l_\mathrm{g}) = \min(\eta l_g, \gamma) \tag{120}$$

A straightforward calculation shows that

$$\frac{1}{2}\min(\eta l_g, \gamma) \leq \eta \frac{l_\mathrm{g}}{1 + \eta l_\mathrm{g}/\gamma} \leq \min(\eta l_g, \gamma) \tag{121}$$

Therefore soft clipping is in fact equivalent to hard clipping up to a constant factor 2 in the step size choice. Thus it's easy to see that our results also hold for Algorithm 3. However, compared to hard clipping, soft clipping has the advantage that the function $h_\mathrm{soft}$ in (119) is smooth while $h_\mathrm{hard}$ in (120) is not, as shown in Figure 2. We also empirically observe that the training curve of soft clipping is more smooth than hard clipping.

## Appendix G  Experimental Details in Section 4

Based on the discussion in Appendix F, we use the soft version of clipping algorithms in all the experiments.

### G.1  CIFAR-10

The CIFAR-10 dataset contains 50k images for training and 10k for testing. All the images are $32 \times 32$ RGB bitmaps. We use the standard ResNet-32 architecture. The total number of parameters is 466,906. For all algorithms, we use mini-batch size 128 and weight decay $5 \times 10^{-4}$. For the baseline algorithm, we use SGD with momentum using learning rate $lr = 1.0$ and momentum factor $\beta = 0.9$. Note that we use the momentum defined in Algorithm 1, which is equivalent to a Pytorch implementation with $lr = 0.1$ and $\beta = 0.9$. We optimize ResNet-32 for 150 epochs, and decrease the learning rate at epoch 80 and epoch 120. For other algorithms, we perform a course grid search for $lr$ an $\gamma$, while keeping all the training strategy the same as SGD. We use 5 random seeds ranging from 2016 to 2020, and the results are similar. The plot in Figure 2 uses the random seed 2020.

### G.2  PTB

The Penn Treebank dataset has a vocabulary of size 10k, and 887k/70k/78k words for training/validation/testing. We use the state-of-the-art AWD-LSTM architecture using hidden size 1150 and embedding size 400. The total number of parameters is 23,941,600. For the baseline algorithm, we follow Merity et al. [2017] who use averaged SGD clipping without momentum using learning rate $lr = 30$ and $\gamma = 7.5$. Note that here $\gamma = 7.5$ means that the gradient norm will be clipped to be no more than 0.25. We use the same dropout rate and regularization hyper-parameters in [Merity et al., 2017]. We train AWD-LSTM for 250 epochs, and averaging is triggered when the validation perplexity stops improving. For other algorithms, we perform a course grid search for $lr$ an $\gamma$, while keeping all the training strategy the same as SGD clipping. We use 5 random seeds ranging from 2016 to 2020, and the results are similar. The plot in Figure 2 uses the random seed 2020.

Figure 3: Experimental results on ImageNet.

### G.3 ImageNet

We also conduct experiments on ImageNet dataset. This dataset contains about 1.28 million training images and 50k validation images with various sizes. We train the standard ResNet-50 architecture on this dataset. The total number of parameters is 25,557,032. We use a batch size of 256 on 8 GPUs and a weight decay of $10^{-4}$. For the baseline algorithm, we choose SGD with learning rate $lr = 1.0$ and momentum $\beta = 0.9$, following Goyal et al. [2017]. Note that we use the momentum defined in Algorithm 1, which is equivalent to a Pytorch implementation with $lr = 0.1$ and $\beta = 0.9$. We train the ResNet-50 for 90 epochs, and decrease the learning rate in epoch 30, epoch 60 and epoch 80. For the other algorithms, we perform a course grid search for $lr$ an $\gamma$, while keeping all the training strategy the same as SGD.

Figure 3 plot the training loss curve and validation accuracy curve on ImageNet. All the algorithms reach a validation accuracy of about 76%. However, all the clipping algorithms train faster than the baseline SGD. Mixed clipping performs the best among the four algorithms.

## Appendix H   Additional experiments in $(L_0, L_1)$-smooth setting using MNIST dataset

In this section, we are aiming to construct an optimization problem which provably satisfies the $(L_0, L_1)$-smoothness condition in this paper rather than the traditional $L$-smoothness condition. We then conduct experiments in both deterministic setting and stochastic setting.

We first cosider a binary classification problem. Suppose a dataset $\mathcal{D}$ contains $n$ samples, denoted as $\{(x_i, y_i)\}_{i=1}^n$, where $x_i$ is a $d$-dimensional input vector and $y_i \in \{-1, +1\}$ is the corresponding label. A discriminant function $f$ with parameter $w, b$ is a mapping from $\mathbb{R}^d$ to $\mathbb{R}$ such that $f_{w,b}(x) = w^T x + b$. We use the empirical error under the exponential loss function (122):

$$L(w, b) = \mathop{\mathbb{E}}_{(x,y) \sim \mathcal{D}} \exp(-y f_{w,b}(x)) = \frac{1}{n} \sum_{i=1}^n \exp(-y_i(w^T x_i + b)) \tag{122}$$

In fact, if the exponential function $\exp(\cdot)$ is replaced by $\log(1 + \exp(\cdot))$, the problem becomes the well-known logistic regression. However, logistic loss has bounded second-order derivative (thus is $L$-smooth), while $\exp(\cdot)$ does not. Furthermore, exponential function is (0,1)-smooth, thus we expect $L(w, b)$ is also $(L_0, L_1)$-smooth for some $L_0, L_1$ (see the following proposition). This is why we use exponential loss here. We point out that such exponential loss is also used in a variety of algorithms, such as boosting (AdaBoost).

When the dataset is linearly separable, parameter $w$ will be driven to infinity through optimization, thus adding some regularization is prevalent in linear classification. We use the following term (123) rather than $L_2$ norm for regularization, in order to be compatible with $L(w, b)$.

$$R_\lambda(w) = \sum_{i=1}^d \left[ \frac{\exp(\lambda w_i) + \exp(-\lambda w_i)}{2} - 1 \right] \tag{123}$$

(a) The deterministic setting　　　　(b) The stochastic setting

Figure 4: Experimental results on MNIST.

In fact, $R_\lambda(w)$ is similar to weight decay regularization in that $R_\lambda(w) = \frac{1}{2}\lambda^2\|w\|^2 + O(\lambda^4\|w\|^4)$ when $w$ is small.

The total loss $E_\lambda(w,b) = L(w,b) + R_\lambda(w)$. We now claim that $E_\lambda(w,b)$ is indeed $(L_0, L_1)$-smooth.

**Proposition H.1** *Assume bias term $b = 0$ for simplicity. Suppose the data points have bounded norm, i.e. $\|x_i\| \le R$ for all $i$ and $\lambda < R$. Let the loss function $E_\lambda(w,0)$ be defined above. Then for every $\rho_1 > 0, \rho_2 > 0$, $\rho = \rho_1 + \rho_2$, $E_\lambda(w,0)$ is $(L_0, L_1)$-smooth w.r.t $w$ for*

$$L_0 = \max\left(\frac{(1+\rho)\sqrt{d}}{\lambda}R^2(R+d\lambda), (R^2 + d\lambda^2)\left(\frac{n(R^2 + d\lambda^2)}{\rho_1 R^2}\right)^{1+\frac{1}{\rho_2}}\right), L_1 = \frac{(1+\rho)\sqrt{d}}{\lambda}R^2.$$

We use MNIST dataset in this section, which contains 60,000 hand-writing training images. We only evaluate the training speed for different algorithms on the training set rather than the generalization capability. The loss functions is defined to be the sum of ten losses, each of which corresponds to the loss of a binary classification problem to recognize number 0 to 9. Regularization coefficient $\lambda$ is set to be 0.02.

To compare different algorithms, we choose the best hyperparameters $lr$ and $\gamma$ for each algorithm based on a *careful* grid search. $\nu$ is set to be 0.7 for mixed clipping. The parameter initalization and all inputs in the schocastic setting are the same for all algorithms. For each run, we average the loss of the last 5 epoch in order to reduce variance. In the deterministic setting we train 500 epochs, each of which uses the entire dataset. In the stochastic setting we train 50 epochs with a mini-batch size 200. We run on 5 different random seeds ranging from 2016 to 2020 altogether and average their results.

Figure 4 plots the results. It is clear that in both settings, clipping is vital to a fast convergence. Also, momentum helps training, and mixed clipping performs the best in the stochastic setting.

## H.1  Proof of Proposition H.1

Consider the augmented dataset $\tilde{\mathcal{D}}$ containing $n + 2d$ data points $\{z_i\}_{i=1}^{n+2d}$, with

$$z_i = \begin{cases} -x_i y_i & i \le n \\ \lambda e_{i-n} & n < i \le n+d \\ -\lambda e_{i-n-d} & n+d < i \le n+2d \end{cases} \tag{124}$$

where $e_i$ is the vector with all zero entries except the $i$th entry which is one. Denote coefficient vector $c \in \mathbb{R}^{n+2d}$ with elements $c_i = 1/n$ if $i \le n$ and $c_i = 1/2$ otherwise. It directly follows that the original problem with regularization term can be written as:

$$E_\lambda(w) = \frac{1}{n}\sum_{i=1}^{n}\exp(w^T z_i) + \frac{1}{2}\sum_{i=n+1}^{n+2d}\exp(w^T z_i) - d = \sum_{i=1}^{n+2d}c_i\exp(w^T z_i) - d \tag{125}$$

Let $M = \max_{i \in [n+2d]} w^T z_i$. Let $\rho_1 > 0, \rho_2 > 0$ be two constants. Pick $M_0 = \left(1 + \frac{1}{\rho_2}\right) \log \frac{n(R^2 + d\lambda^2)}{\rho_1 R^2}$.

We consider the following two cases:

(1)$M \leq M_0$. In this case $\|\nabla^2 E(w)\|$ can be directly upper bounded:

$$
\begin{aligned}
\|\nabla^2 E(w)\| &\leq \frac{1}{n} \sum_{i=1}^{n} \exp(w^T z_i) \|z_i\|^2 + \frac{1}{2} \sum_{i=n+1}^{n+2d} \exp(w^T z_i) \|z_i\|^2 \\
&\leq (R^2 + d\lambda^2) \exp(M) \\
&\leq (R^2 + d\lambda^2) \left(\frac{n(R^2 + d\lambda^2)}{\rho_1 R^2}\right)^{1 + \frac{1}{\rho_2}}
\end{aligned}
\tag{126}
$$

The first inequality in (126) uses the triangular inequality of matrix spectral norm and $\|z z^T\| = \|z^T z\| = \|z\|^2$.

(2)$M > M_0$. Decompose $M_0$ to be $M_0 = M_1 + M_2$ where

$$
M_1 = \log \frac{n(R^2 + d\lambda^2)}{\rho_1 R^2}, M_2 = \frac{1}{\rho_2} \log \frac{n(R^2 + d\lambda^2)}{\rho_1 R^2}.
$$

Define set $I = \{i \in [n+2d] : w^T z_i \geq M - M_1\}$ and $I_2 = \{i \in [n+2d] : w^T z_i < 0\}$. Then

$$
\|\nabla E(w)\| = \sum_{i=1}^{n+2d} c_i \exp(w^T z_i) z_i \tag{127}
$$

$$
\geq \sum_{i=1}^{n+2d} c_i \exp(w^T z_i) \frac{w^T z_i}{\|w\|} \tag{128}
$$

$$
\geq \sum_{i \in I} c_i \exp(w^T z_i) \frac{M - M_1}{\|w\|} - \sum_{i \in I_2} c_i \|z_i\| \tag{129}
$$

$$
\geq \sum_{i \in I} c_i \exp(w^T z_i) \frac{M - M_1}{\|w\|} - (R + d\lambda) \tag{130}
$$

In (128) we use the Cauchy-Schwartz inequality; In (129) we partition the index $\{i : i \in [n+2d]\}$ to three subsets $I$, $I_2$ and $[n+2d] \backslash (I \cup I_2)$, and use the lower bound and upper bound of $w^T z_i > 0$ for each set.

Similar, we can upper bound $\|\nabla^2 E(w)\|$:

$$
\|\nabla^2 E(w)\| \leq \sum_{i \in I} c_i \exp(w^T z_i) \|z_i\|^2 + \sum_{i \notin I} c_i \exp(w^T z_i) \|z_i\|^2 \tag{131}
$$

$$
\leq \sum_{i \in I} c_i \exp(w^T z_i) R^2 + (R^2 + d\lambda^2) \exp(M - M_1) \tag{132}
$$

To bound $\exp(M_1)$, we again bound $\|\nabla E(w)\|$ from a different perspective:

$$
\|\nabla E(w)\| \geq \sum_{i \in I} c_i \exp(w^T z_i) \frac{w^T z_i}{\|w\|} - \sum_{i \in I_2} c_i \|z_i\| \tag{133}
$$

$$
\geq \frac{1}{n} \exp(M) \frac{M}{\|w\|} - (R + d\lambda) \tag{134}
$$

where (134) is obtained by selecting the $i$ with the largest $w^T z_i$ which is equal to $M$. Substitute (130) and (134) into (132) then we get

$$
\|\nabla^2 E(w)\| \leq \left(\frac{R^2}{M - M_1} + \frac{n(R^2 + d\lambda^2)}{M \exp(M_1)}\right) \|w\|(\|\nabla E(w)\| + R + d\lambda) \tag{135}
$$

$$
= \left(\frac{R^2}{M - M_1} + \frac{\rho_1 R^2}{M}\right) \|w\|(\|\nabla E(w)\| + R + d\lambda) \tag{136}
$$

Since $M = \max\limits_{i \in [n+2d]} w^T z_i$ implies that $|\lambda w_k| \leq M$ for all $k \in [d]$ from (124), we can upper bound the norm of $w$: $\|w\| \leq \frac{M\sqrt{d}}{\lambda}$. Substitute this into (136) we get

$$\|\nabla^2 E(w)\| \leq \left(\frac{M}{M - M_1} + \rho_1\right) \frac{\sqrt{d}}{\lambda} R^2 (\|\nabla E(w)\| + R + d\lambda) \tag{137}$$

$$\leq \frac{(1 + \rho_1 + \rho_2)\sqrt{d}}{\lambda} R^2 (\|\nabla E(w)\| + R + d\lambda) \tag{138}$$

Combining the above two cases concludes the proof.