[Reviews · NeurIPS 2020]

Review 1

Summary and Contributions: Provides a new analysis with a clearer dependence on key problem constants then existing approaches to the analysis of clipped gradient descent with and without momentum.

Strengths: The analysis performed in this paper seems like a good contribution. I've not seen the Lyapunov function they use before, the use of a non-squared norm is unusual and interesting. The interplay between the constants also provides insight into what's going on, the fact that the convergence rate is dominated by the L_0 constant matches intuition, as well as the pragmatic fact that gradient clipping is most useful at the earliest iterations. Analyzing methods with step-dependence scaling in the stochastic case is challenging, and so the contribution in this respect is strong. The experiments in the paper are good, they perform a hyper parameter search and the plots are clear and easy to read.

Weaknesses: A more in-depth discussion of the mixed method which appears to work better empirically may improve the paper. I would move the imagenet experiments into the main body of the paper as well, as having three datasets in the main body of the paper is the informal standard. The final test errors should be reported in each case, with the test error shown. Ideally 10 trials instead of 5 should be used. Is there a reason to use "energy function" instead of "Lyapunov function"? The later is more standard.

Correctness: I have not checked the proofs in the appendix.

Clarity: This paper is very clear and concise.

Relation to Prior Work: There appears to be a reasonable set of references to existing research.

Reproducibility: Yes

Additional Feedback: UPDATE: I have discussed the paper with other reviewers.


Review 2

Summary and Contributions: The authors improve on an analyis by Zhang et al. 2020 which introduced the notion of (L0,l1) smooth functions and derived bounds on clipped GD and SGD. The improvements contributed by this work are: - a more general framework of analysis which covers the work of Zhang et al, but also allows for the consideration of clipped/normalized momentum as well as "mixed clipping", a less commonly used variant which yielded good empirical results - obtaining tight bounds for this general framework, yielding much sharper rates than the original work - experimentally verifying the theoretical results

Strengths: The paper gets nice empirical results with a relatively simple analytical framework which seems to yield good bounds. The fact that the proof is based on an energy function means that it can handle momentum and the temporary increases in function value as well and so could possibly be used as a stepping stone to also analyse the behaviour of adaptive algorithms under clipping (although I have to couch this assessment since I am not very familiar with the theoretical works in this area).

Weaknesses: - small and easily fixable: I think the experiment section would gain additional context if there was a comparison with AdaM with default parameters included, since it's the current "default" optimiser and could give context for the performance reached by SGD. In particular, since the work [https://arxiv.org/abs/1705.08292](https://arxiv.org/abs/1705.08292) showed the tradeoffs involved between training speed of of adaptive methods and performance at test time, seeing whether the proposed method retains the improved generalization while benefiting from a speedup using momentum would be quite interesting Edit: authors addresed this in rebuttal - in order to compare the results of this work with those of Zhang et al. 2020, using the same resnet architecture+hyperparameters for them for the CIFAR10 evaluation and reporting the validation loss on top of perplexity (while also matching hyperparameters) would be nice Edit: fixed in rebuttal - also, adding a run of the algorithms analysed in Zhang et al. for direct comparison of empirical behaviour to see the impact of momentum and mixed clipping - reporting standard deviation as shaded area in the plots would be appreciated - some discussion on what the theory implies for the values of the clipping parameter gamma, the step size and the momentum parameters given specific L0,L1 of the functions (+ the sigma for the stochastic gradient oracle) and some constructed toy examples to verify predictive power of the framework would be a nice Edit: the rebuttal did not comment on the request for standard deviations across multiple training runs, which I would stress on adding, otherwise I feel most of these were adressed

Correctness: I have not been able to go through the proofs in depth, but the general arguments seem to make sense to me. The empirical methodology seems sound, although reporting the standard deviation of the runs would be appreciated

Clarity: The paper is well written and clear to me, despite not being an expert I felt I could follow the general shape of the proofs.

Relation to Prior Work: The paper builds heavily on Zhang et al 2020 and does adequate comparisons with it and other relevant work as best as I can judge, but the experimental section could use some more direct comparisons.

Reproducibility: Yes

Additional Feedback: The broader impact discussion is well done, although the privacy comment seems a bit thrown in. Another (half) sentence how momentum/other gradient information can leak training data information and in what settings would help clear this up


Review 3

Summary and Contributions: This paper present a new analysis of (momentum) SGD with gradient clipping. Its contributions are two fold: (1) improve the rate compared with a recent work; (2) include more variants in the framework.

Strengths: The strengthen of this paper lies at the theoretical improvement in terms of the problem parameter L1. It improves our understanding of SGD with gradient clipping.

Weaknesses: Although the improvement is interesting, it has some drawbacks in analysis: 1. It is unfair to compare with Ghadimi and Lan 2013, because they assume weaker assumptions about stochastic gradient, which is very important in the context of gradient clipping. The reason that gradient clipping is useful is that stochastic gradient might have large error, it could be heavy-tail or unbounded. However, this work simply assume that the difference between stochastic gradient and true gradient is bounded, which implies that stochastic gradient is bounded if we assume the true gradient is bounded. This is very problematic. Even one can relax the assumption to sub-Gaussian light tail assumption, it is still not valid in practice. 2. The theoretical version of the proposed algorithm is very impractical. In particular, the clipping parameter is very small in the level of \epsilon^2, and the step size \eta is also very small. This makes a huge gap between theory and experiments, where \gamma is set to be a constant. 3. The theoretical results seems indicate the setting \beta=0 gives the fastest convergence. This seems problematic. There should be some tradeoff in the iteration complexity involving \beta.

Correctness: The small value of \gamma seems problematic and its gap with the empirical study exists.

Clarity: Yes

Relation to Prior Work: Yes

Reproducibility: Yes

Additional Feedback:

[Author Response · NeurIPS 2020]

We would like to thank the entire review team for their efforts and comments. In particular, we would like to thank
Reviewer 1 and 2 for the positive comments and Reviewer 3 for sharing the concerns.

**To Reviewer 1** We would like to thank the suggestions from Reviewer 1. We will follow these suggestions in the
revised version. Specifically, we will move the imagenet experiments to the main text and repeat the experiments in
10 times. We will change the term "energy funcion" to "Lyapunov function", and add more discussions of the mixed
clipping method. The test errors/validation perplexities of all the algorithms are shown in the following table.

|          | CIFAR-10 test acc | PTB validation ppl | ImageNet validation top1 acc |
|----------|-------------------|--------------------|------------------------------|
| SGD      | 93.0              | 68.87              | 76.1                         |
| SGD Clip | 93.3              | 63.25              | 75.9                         |
| Mom Clip | 93.2              | 63.05              | 76.1                         |
| Mix Clip | 93.2              | 62.17              | 76.1                         |

**To Reviewer 2** We would like to thank the suggestions from Reviewer 2. We have added experiments using Adam
optimizer with best hyper-parameters (see Figures (a, b)). Results show that the training speed using Adam is faster
than using baseline SGD. However, Adam generalizes worse. We also add experiments using the same ResNet archi-
tecture (ResNet20) and the same hyper-parameters as Zhang et al. [2020] (see Figure (c)). All algorithms can achieve
95% test accuracy (as reported in Zhang et al. [2020]), and the training curve is similar to Figure (a). For PTB dataset,
the validation loss is 4.13 using mixed clipping. We will plot standard deviation as shaded area in the revised version
of our paper. We turn the step size and clipping hyper-parameters by grid-search.

(a) CIFAR-10      (b) PTB      (c) CIFAR10 (ResNet20)

**To Reviewer 3 about the noise assumption** We would like to thank Reviewer 3 for raising this concern. We justify
the reasonability of our assumption below:

(A) Our paper follows the research line typically from Zhang et al. [2020]. This research line attempts to understand
the strength of clipping methods for non-smooth (in the traditional sense) objective functions. Note that Zhang et al.
[2020] have made the *same* assumptions, in that they also assume the noise is bounded. We improve their complexity
under the same conditions.

(B) We are aware that Ghadimi and Lan [2013] obtains the upper bound complexity under a weaker assumption.
However, shown in Section 3.3 (line 234), there is a hard objective function that satisfies our (stronger) assumption, for
which the Stochastic Gradient Descent algorithm must take $\Omega\left(\Delta L\sigma^2\epsilon^{-4}\right)$ to find a first-order stationary point. Hence
Ghadimi's result *cannot* be further improved under our (stronger) assumption. In contrast, we show that clipping
methods enjoy a better complexity of $\mathcal{O}\left(\Delta L_0\sigma^2\epsilon^{-4}\right)$.

(C) We think that the bounded (or Sub-Gaussian tail) noise assumption is quite *common* in the non-convex stochas-
tic optimization field. Many analyses adopt this assumption to prove convergence, especially for adaptive gradient
methods (e.g. Li and Orabona [2019]) and escaping saddle points (e.g. Fang et al. [2018]).

**To Reviewer 3 about the clipping parameters and step sizes.** Sorry for the confusion about clipping parameters.
Taking Theorem 3.2 as an example. The clipping threshold of gradient norm is actually $\gamma/\eta = 5\sigma = \Theta(1)$, which is
*at constant magnitude* and independent of $\epsilon$. It is true that the step size provided in this result is $\mathcal{O}(\epsilon^2)$. This step size
is common in the analyses of non-convex stochastic algorithms (e.g. Ghadimi and Lan [2013]). In practice, we may
choose a relatively large step size at beginning and gradually decrease it. Our analysis can also be extended to this
setting. However, in the final state, the step size still needs to be $\mathcal{O}(\epsilon^2)$.

**To Reviewer 3 about the best choice of $\beta$.** Theorem 3.2 implies that for sufficiently small $\epsilon$, the number of iterations
needed to reach an $\epsilon$-stationary point *does not* depend on $\beta$.

[Meta-Review · NeurIPS 2020]

Two reviewers indicate acceptance, and one reviewer indicates reject. The main concern of that reviewer is the unrealistic assumption on the bounded noise (or sub-Gaussian tail). Reviewer 1 indicates that this assumption is common and thus acceptable, but also points out it is reasonable to expect that for clipping algorithm the assumption can be relaxed. R1 and R2 pointed out the Lyapunov function seems to be novel, and the analysis is non-trivial. Therefore, I recommend accept. That being said, I agree that this is a valid concern, and suggest the authors to add more discussions on the limitation of this assumption, and also point out that relaxing the assumption for clipping algorithms is more important/urgent than general cases.